# Unlabeled Data Improves Adversarial Robustness

**Yair Carmon**[*]
Stanford University
yairc@stanford.edu

**Aditi Raghunathan**[*]
Stanford University
aditir@stanford.edu

**Ludwig Schmidt**
UC Berkeley
ludwig@berkeley.edu

**Percy Liang**
Stanford University
pliang@cs.stanford.edu

**John C. Duchi**
Stanford University
jduchi@stanford.edu

## Abstract

We demonstrate, theoretically and empirically, that adversarial robustness can significantly benefit from semisupervised learning. Theoretically, we revisit the simple Gaussian model of Schmidt et al. [41] that shows a sample complexity gap between standard and robust classification. We prove that unlabeled data bridges this gap: a simple semisupervised learning procedure (self-training) achieves high robust accuracy using the same number of labels required for achieving high standard accuracy. Empirically, we augment CIFAR-10 with 500K unlabeled images sourced from 80 Million Tiny Images and use robust self-training to outperform state-of-the-art robust accuracies by over 5 points in (i) $\ell_\infty$ robustness against several strong attacks via adversarial training and (ii) certified $\ell_2$ and $\ell_\infty$ robustness via randomized smoothing. On SVHN, adding the dataset's own extra training set with the labels removed provides gains of 4 to 10 points, within 1 point of the gain from using the extra labels.

## 1 Introduction

The past few years have seen an intense research interest in making models robust to adversarial examples [44, 4, 3]. Yet despite a wide range of proposed defenses, the state-of-the-art in adversarial robustness is far from satisfactory. Recent work points towards sample complexity as a possible reason for the small gains in robustness: Schmidt et al. [41] show that in a simple model, learning a classifier with non-trivial adversarially robust accuracy requires substantially more samples than achieving good "standard" accuracy. Furthermore, recent empirical work obtains promising gains in robustness via transfer learning of a robust classifier from a larger labeled dataset [18]. While both theory and experiments suggest that more training data leads to greater robustness, following this suggestion can be difficult due to the cost of gathering additional data and especially obtaining high-quality labels.

To alleviate the need for carefully labeled data, in this paper we study adversarial robustness through the lens of semisupervised learning. Our approach is motivated by two basic observations. First, adversarial robustness essentially asks that predictors be stable around naturally occurring inputs. Learning to satisfy such a stability constraint should not inherently require labels. Second, the added requirement of robustness fundamentally alters the regime where semi-supervision is useful. Prior work on semisupervised learning mostly focuses on improving the standard accuracy by leveraging

---

[*] Equal contribution.
Code and data are available on GitHub at `https://github.com/yaircarmon/semisup-adv` and on CodaLab at `https://bit.ly/349WsAC`.

unlabeled data. However, in our adversarial setting the labeled data alone already produce accurate (but not robust) classifiers. We can use such classifiers on the unlabeled data and obtain useful *pseudo-labels*, which directly suggests the use of *self-training*—one of the oldest frameworks for semisupervised learning [42, 8], which applies a supervised training method on the pseudo-labeled data. We provide theoretical and experimental evidence that self-training is effective for adversarial robustness.

The first part of our paper is theoretical and considers the simple $d$-dimensional Gaussian model [41] with $\ell_\infty$-perturbations of magnitude $\epsilon$. We scale the model so that $n_0$ labeled examples allow for learning a classifier with nontrivial standard accuracy, and roughly $n_0 \cdot \epsilon^2 \sqrt{d/n_0}$ examples are necessary for attaining any nontrivial robust accuracy. This implies a sample complexity gap in the high-dimensional regime $d \gg n_0 \epsilon^{-4}$. In this regime, we prove that self training with $O(n_0 \cdot \epsilon^2 \sqrt{d/n_0})$ unlabeled data and just $n_0$ labels achieves high robust accuracy. Our analysis provides a refined perspective on the sample complexity barrier in this model: the increased sample requirement is exclusively on unlabeled data.

Our theoretical findings motivate the second, empirical part of our paper, where we test the effect of unlabeled data and self-training on standard adversarial robustness benchmarks. We propose and experiment with robust self-training (RST), a natural extension of self-training that uses standard supervised training to obtain pseudo-labels and then feeds the pseudo-labeled data into a supervised training algorithm that targets adversarial robustness. We use TRADES [56] for *heuristic* $\ell_\infty$-robustness, and stability training [57] combined with randomized smoothing [9] for *certified* $\ell_2$-robustness.

For CIFAR-10 [22], we obtain 500K unlabeled images by mining the 80 Million Tiny Images dataset [46] with an image classifier. Using RST on the CIFAR-10 training set augmented with the additional unlabeled data, we outperform state-of-the-art *heuristic* $\ell_\infty$-robustness against strong iterative attacks by 7%. In terms of *certified* $\ell_2$-robustness, RST outperforms our fully supervised baseline by 5% and beats previous state-of-the-art numbers by 10%. Finally, we also match the state-of-the-art certified $\ell_\infty$-robustness, while improving on the corresponding standard accuracy by over 16%. We show that some natural alternatives such as virtual adversarial training [30] and aggressive data augmentation do not perform as well as RST. We also study the sensitivity of RST to varying data volume and relevance.

Experiments with SVHN show similar gains in robustness with RST on semisupervised data. Here, we apply RST by removing the labels from the 531K extra training data and see 4–10% increases in robust accuracies compared to the baseline that only uses the labeled 73K training set. Swapping the pseudo-labels for the true SVHN extra labels increases these accuracies by at most 1%. This confirms that the majority of the benefit from extra data comes from the inputs and not the labels.

In independent and concurrent work, Uesato et al. [48], Najafi et al. [32] and Zhai et al. [55] also explore semisupervised learning for adversarial robustness. See Section 6 for a comparison.

Before proceeding to the details of our theoretical results in Section 3, we briefly introduce relevant background in Section 2. Sections 4 and 5 then describe our adversarial self-training approach and provide comprehensive experiments on CIFAR-10 and SVHN. We survey related work in Section 6 and conclude in Section 7.

## 2  Setup

**Semi-supervised classification task.**  We consider the task of mapping input $x \in \mathcal{X} \subseteq \mathbb{R}^d$ to label $y \in \mathcal{Y}$. Let $P_{x,y}$ denote the underlying distribution of $(x,y)$ pairs, and let $P_x$ denote its marginal on $\mathcal{X}$. Given training data consisting of (i) labeled examples $(X,Y) = (x_1,y_1),...(x_n,y_n) \sim P_{x,y}$ and (ii) unlabeled examples $\tilde{X} = \tilde{x}_1, \tilde{x}_2,...\tilde{x}_{\tilde{n}} \sim P_x$, the goal is to learn a classifier $f_\theta : \mathcal{X} \to \mathcal{Y}$ in a model family parameterized by $\theta \in \Theta$.

**Error metrics.**  The standard quality metric for classifier $f_\theta$ is its error probability,

$$\text{err}_{\text{standard}}(f_\theta) := \mathbb{P}_{(x,y) \sim P_{x,y}} \big(f_\theta(x) \neq y\big). \tag{1}$$

We also evaluate classifiers on their performance on *adversarially perturbed inputs*. In this work, we consider perturbations in a $\ell_p$ norm ball of radius $\epsilon$ around the input, and define the corresponding

robust error probability,

$$\mathrm{err}_{\mathrm{robust}}^{p,\epsilon}(f_\theta) := \mathbb{P}_{(x,y) \sim P_{\mathsf{x},\mathsf{y}}}\left(\exists x' \in \mathcal{B}_\epsilon^p(x), f_\theta(x') \neq y\right) \text{ for } \mathcal{B}_\epsilon^p(x) := \{x' \in \mathcal{X} \,|\, \|x'-x\|_p \leq \epsilon\}. \quad (2)$$

In this paper we study $p=2$ and $p=\infty$. We say that a classifier $f_\theta$ has *certified* $\ell_p$ accuracy $\xi$ when we can *prove* that $\mathrm{err}_{\mathrm{robust}}^{p,\epsilon}(f_\theta) \leq 1-\xi$.

**Self-training.** Consider a supervised learning algorithm A that maps a dataset $(X,Y)$ to parameter $\theta$. *Self-training* is the straightforward extension of A to a semisupervised setting, and consists of the following two steps. First, obtain an intermediate model $\hat{\theta}_{\mathrm{intermediate}} = \mathsf{A}(X,Y)$, and use it to generate *pseudo-labels* $\tilde{y}_i = f_{\hat{\theta}_{\mathrm{intermediate}}}(\tilde{x}_i)$ for $i \in [\tilde{n}]$. Second, combine the data and pseudo-labels to obtain a final model $\hat{\theta}_{\mathrm{final}} = \mathsf{A}([X,\tilde{X}],[Y,\tilde{Y}])$.

## 3 Theoretical results

In this section, we consider a simple high-dimensional model studied in [41], which is the only known formal example of an information-theoretic sample complexity gap between standard and robust classification. For this model, we demonstrate the value of unlabeled data—a simple self-training procedure achieves high robust accuracy, when achieving non-trivial robust accuracy using the labeled data alone is impossible.

**Gaussian model.** We consider a binary classification task where $\mathcal{X} = \mathbb{R}^d$, $\mathcal{Y} = \{-1,1\}$, $y$ uniform on $\mathcal{Y}$ and $x|y \sim \mathcal{N}(y\mu, \sigma^2 I)$ for a vector $\mu \in \mathbb{R}^d$ and coordinate noise variance $\sigma^2 > 0$. We are interested in the standard error (1) and robust error $\mathrm{err}_{\mathrm{robust}}^{\infty,\epsilon}$ (2) for $\ell_\infty$ perturbations of size $\epsilon$.

**Parameter setting.** We choose the model parameters to meet the following desiderata: (i) there exists a classifier that achieves very high robust and standard accuracies, (ii) using $n_0$ examples we can learn a classifier with non-trivial standard accuracy and (iii) we require much more than $n_0$ examples to learn a classifier with nontrivial robust accuracy. As shown in [41], the following parameter setting meets the desiderata,

$$\epsilon \in (0,\tfrac{1}{2}), \ \ \|\mu\|^2 = d \ \text{ and } \ \frac{\|\mu\|^2}{\sigma^2} = \sqrt{\frac{d}{n_0}} \gg \frac{1}{\epsilon^2}. \quad (3)$$

When interpreting this setting it is useful to think of $\epsilon$ as fixed and of $d/n_0$ as a large number, i.e. a highly overparameterized regime.

### 3.1 Supervised learning in the Gaussian model

We briefly recapitulate the sample complexity gap described in [41] for the fully supervised setting.

**Learning a simple linear classifier.** We consider linear classifiers of the form $f_\theta = \mathrm{sign}(\theta^\top x)$. Given $n$ labeled data $(x_1,y_1),...,(x_n,y_n) \overset{\mathrm{iid}}{\sim} P_{\mathsf{x},\mathsf{y}}$, we form the following simple classifier

$$\hat{\theta}_n := \frac{1}{n}\sum_{i=1}^{n} y_i x_i. \quad (4)$$

We achieve nontrivial standard accuracy using $n_0$ examples; see Appendix A.2 for proof of the following (as well as detailed rates of convergence).

**Proposition 1.** *There exists a universal constant $r$ such that for all $\epsilon^2 \sqrt{d/n_0} \geq r$,*

$$n \geq n_0 \Rightarrow \mathbb{E}_{\hat{\theta}_n} \mathrm{err}_{\mathrm{standard}}\left(f_{\hat{\theta}_n}\right) \leq \frac{1}{3} \ \text{ and } \ n \geq n_0 \cdot 4\epsilon^2\sqrt{\frac{d}{n_0}} \Rightarrow \mathbb{E}_{\hat{\theta}_n} \mathrm{err}_{\mathrm{robust}}^{\infty,\epsilon}\left(f_{\hat{\theta}_n}\right) \leq 10^{-3}.$$

Moreover, as the following theorem states, no learning algorithm can produce a classifier with nontrivial robust error without observing $\widetilde{\Omega}(n_0 \cdot \epsilon^2 \sqrt{d/n_0})$ examples. Thus, a sample complexity gap forms as $d$ grows.

**Theorem 1** ([41]). *Let $\mathsf{A}_n$ be any learning rule mapping a dataset $S \in (\mathcal{X} \times \mathcal{Y})^n$ to classifier $\mathsf{A}_n[S]$. Then,*

$$n \leq n_0 \frac{\epsilon^2\sqrt{d/n_0}}{8\log d} \Rightarrow \mathbb{E}\,\mathrm{err}_{\mathrm{robust}}^{\infty,\epsilon}(\mathsf{A}_n[S]) \geq \frac{1}{2}(1-d^{-1}), \quad (5)$$

*where the expectation is with respect to the random draw of $S \sim P_{\mathsf{x},\mathsf{y}}^n$ as well as possible randomization in $\mathsf{A}_n$.*

## 3.2 Semi-supervised learning in the Gaussian model

We now consider the semisupervised setting with $n$ labeled examples and $\tilde{n}$ additional unlabeled examples. We apply the self-training methodology described in Section 2 on the simple learning rule (4); our intermediate classifier is $\hat{\theta}_{\text{intermediate}} := \hat{\theta}_n = \frac{1}{n}\sum_{i=1}^{n} y_i x_i$, and we generate pseudo-labels $\tilde{y}_i := f_{\hat{\theta}_{\text{intermediate}}}(\tilde{x}_i) = \text{sign}(\tilde{x}_i^\top \hat{\theta}_{\text{intermediate}})$ for $i=1,...,\tilde{n}$. We then learning rule (4) to obtain our final semisupervised classifier $\hat{\theta}_{\text{final}} := \frac{1}{\tilde{n}}\sum_{i=1}^{\tilde{n}} \tilde{y}_i \tilde{x}_i$. The following theorem guarantees that $\hat{\theta}_{\text{final}}$ achieves high robust accuracy.

**Theorem 2.** *There exists a universal constant $\tilde{r}$ such that for $\epsilon^2 \sqrt{d/n_0} \geq \tilde{r}$, $n \geq n_0$ labeled data and additional $\tilde{n}$ unlabeled data,*

$$\tilde{n} \geq n_0 \cdot 288\epsilon^2 \sqrt{\frac{d}{n_0}} \Rightarrow \mathbb{E}_{\hat{\theta}_{\text{final}}} \text{err}_{\text{robust}}^{\infty,\epsilon}\left(f_{\hat{\theta}_{\text{final}}}\right) \leq 10^{-3}.$$

Therefore, compared to the fully supervised case, the self-training classifier requires only a constant factor more input examples, and roughly a factor $\epsilon^2 \sqrt{d/n_0}$ fewer labels. We prove Theorem 2 in Appendix A.4, where we also precisely characterize the rates of convergence of the robust error; the outline of our argument is as follows. We have $\hat{\theta}_{\text{final}} = (\frac{1}{\tilde{n}}\sum_{i=1}^{\tilde{n}} \tilde{y}_i y_i)\mu + \frac{1}{\tilde{n}}\sum_{i=1}^{\tilde{n}} \tilde{y}_i \varepsilon_i$ where $\varepsilon_i \sim \mathcal{N}(0,\sigma^2 I)$ is the noise in example $i$. We show (in Appendix A.4) that with high probability $\frac{1}{\tilde{n}}\sum_{i=1}^{\tilde{n}} \tilde{y}_i y_i \geq \frac{1}{6}$ while the variance of $\frac{1}{\tilde{n}}\sum_{i=1}^{\tilde{n}} \tilde{y}_i \varepsilon_i$ goes to zero as $\tilde{n}$ grows, and therefore the angle between $\hat{\theta}_{\text{final}}$ and $\mu$ goes to zero. Substituting into a closed-form expression for $\text{err}_{\text{robust}}^{\infty,\epsilon}(f_{\hat{\theta}_{\text{final}}})$ (Eq. (11) in Appendix A.1) gives the desired upper bound. We remark that other learning techniques, such as EM and PCA, can also leverage unlabeled data in this model. The self-training procedure we describe is similar to 2 steps of EM [11].

## 3.3 Semisupervised learning with irrelevant unlabeled data

In Appendix A.5 we study a setting where only $\alpha\tilde{n}$ of the unlabeled data are relevant to the task, where we model the relevant data as before, and the irrelevant data as having no signal component, i.e., with $y$ uniform on $\{-1,1\}$ and $x \sim \mathcal{N}(0,\sigma^2 I)$ independent of $y$. We show that for any fixed $\alpha$, high robust accuracy is still possible, but the required number of *relevant* examples grows by a factor of $1/\alpha$ compared to the amount of unlabeled examples require to achieve the same robust accuracy when all the data is relevant. This demonstrates that irrelevant data can significantly hinder self-training, but does not stop it completely.

## 4 Semi-supervised learning of robust neural networks

Existing adversarially robust training methods are designed for the supervised setting. In this section, we use these methods to leverage additional unlabeled data by adapting the self-training framework described in Section 2.

---
**Meta-Algorithm 1** Robust self-training
---
    **Input:** Labeled data $(x_1,y_1,...,x_n,y_n)$ and unlabeled data $(\tilde{x}_1,...,\tilde{x}_{\tilde{n}})$

    **Parameters:** Standard loss $L_{\text{standard}}$, robust loss $L_{\text{robust}}$ and unlabeled weight $w$

1: Learn $\hat{\theta}_{\text{intermediate}}$ by minimizing $\sum_{i=1}^{n} L_{\text{standard}}(\theta,x_i,y_i)$

2: Generate pseudo-labels $\tilde{y}_i = f_{\hat{\theta}_{\text{intermediate}}}(\tilde{x}_i)$ for $i=1,2,...\tilde{n}$

3: Learn $\hat{\theta}_{\text{final}}$ by minimizing $\sum_{i=1}^{n} L_{\text{robust}}(\theta,x_i,y_i) + w\sum_{i=1}^{\tilde{n}} L_{\text{robust}}(\theta,\tilde{x}_i,\tilde{y}_i)$

---

Meta-Algorithm 1 summarizes robust-self training. In contrast to standard self-training, we use a different supervised learning method in each stage, since the intermediate and the final classifiers have different goals. In particular, the only goal of $\hat{\theta}_{\text{intermediate}}$ is to generate high quality pseudo-labels for the (non-adversarial) unlabeled data. Therefore, we perform standard training in the first stage, and robust training in the second. The hyperparameter $w$ allows us to upweight the labeled data, which in some cases may be more relevant to the task (e.g., when the unlabeled data comes form a different distribution), and will usually have more accurate labels.

### 4.1 Instantiating robust self-training

Both stages of robust self-training perform supervised learning, allowing us to borrow ideas from the literature on supervised standard and robust training. We consider neural networks of the form $f_\theta(x) = \text{argmax}_{y \in \mathcal{Y}} p_\theta(y \mid x)$, where $p_\theta(\cdot \mid x)$ is a probability distribution over the class labels.

**Standard loss.** As in common, we use the multi-class logarithmic loss for standard supervised learning,
$$L_{\text{standard}}(\theta, x, y) = -\log p_\theta(y \mid x).$$

**Robust loss.** For the supervised robust loss, we use a robustness-promoting regularization term proposed in [56] and closely related to earlier proposals in [57, 30, 20]. The robust loss is
$$L_{\text{robust}}(\theta, x, y) = L_{\text{standard}}(\theta, x, y) + \beta L_{\text{reg}}(\theta, x), \tag{6}$$
$$\text{where } L_{\text{reg}}(\theta, x) := \max_{x' \in \mathcal{B}_\epsilon^p(x)} D_{\text{KL}}(p_\theta(\cdot \mid x) \,\|\, p_\theta(\cdot \mid x')).$$

The regularization term[2] $L_{\text{reg}}$ forces predictions to remain stable within $\mathcal{B}_\epsilon^p(x)$, and the hyperparameter $\beta$ balances the robustness and accuracy objectives. We consider two approximations for the maximization in $L_{\text{reg}}$.

1. **Adversarial training: a heuristic defense via approximate maximization.**

   We focus on $\ell_\infty$ perturbations and use the projected gradient method to approximate the regularization term of (6),
   $$L_{\text{reg}}^{\text{adv}}(\theta, x) := D_{\text{KL}}(p_\theta(\cdot \mid x) \,\|\, p_\theta(\cdot \mid x'_{\text{PG}}[x])), \tag{7}$$
   where $x'_{\text{PG}}[x]$ is obtained via projected gradient ascent on $r(x') = D_{\text{KL}}(p_\theta(\cdot \mid x) \,\|\, p_\theta(\cdot \mid x'))$. Empirically, performing approximate maximization during training is effective in finding classifiers that are robust to a wide range of attacks [29].

2. **Stability training: a certified $\ell_2$ defense via randomized smoothing.**

   Alternatively, we consider stability training [57, 26], where we replace maximization over small perturbations with much larger additive random noise drawn from $\mathcal{N}(0, \sigma^2 I)$,
   $$L_{\text{reg}}^{\text{stab}}(\theta, x) := \mathbb{E}_{x' \sim \mathcal{N}(x, \sigma^2 I)} D_{\text{KL}}(p_\theta(\cdot \mid x) \,\|\, p_\theta(\cdot \mid x')). \tag{8}$$

   Let $f_\theta$ be the classifier obtained by minimizing $L_{\text{standard}} + \beta L_{\text{robust}}^{\text{stab}}$. At test time, we use the following *smoothed* classifier.
   $$g_\theta(x) := \text{argmax}_{y \in \mathcal{Y}} q_\theta(y \mid x), \text{ where } q_\theta(y \mid x) := \mathbb{P}_{x' \sim \mathcal{N}(x, \sigma^2 I)}(f_\theta(x') = y). \tag{9}$$

   Improving on previous work [24, 26], Cohen et al. [9] prove that robustness of $f_\theta$ to large random perturbations (the goal of stability training) implies *certified* $\ell_2$ adversarial robustness of the smoothed classifier $g_\theta$.

## 5 Experiments

In this section, we empirically evaluate robust self-training (RST) and show that it leads to *consistent and substantial* improvement in robust accuracy, on both CIFAR-10 [22] and SVHN [53] and with both adversarial (RST$_{\text{adv}}$) and stability training (RST$_{\text{stab}}$). For CIFAR-10, we mine unlabeled data from 80 Million Tiny Images and study in depth the strengths and limitations of RST. For SVHN, we simulate unlabeled data by removing labels and show that with RST the harm of removing the labels is small. This indicates that most of the gain comes from additional inputs rather than additional labels. Our experiments build on open source code from [56, 9]; we release our data and code at `https://github.com/yaircarmon/semisup-adv` and on CodaLab at `https://bit.ly/349WsAC`.

**Evaluating heuristic defenses.** We evaluate RST$_{\text{adv}}$ and other heuristic defenses on their performance against the strongest known $\ell_\infty$ attacks, namely the projected gradient method [29], denoted PG and the Carlini-Wagner attack [7] denoted CW.

| Model | $\text{PG}_{\texttt{Madry}}$ | $\text{PG}_{\texttt{TRADES}}$ | $\text{PG}_{\texttt{Ours}}$ | CW [7] | Best attack | No attack |
|---|---|---|---|---|---|---|
| $\text{RST}_{\texttt{adv}}$(50K+500K) | 63.1 | 63.1 | 62.5 | 64.9 | **62.5** $\pm$0.1 | **89.7** $\pm$0.1 |
| TRADES [56] | 55.8 | 56.6 | 55.4 | 65.0 | 55.4 | 84.9 |
| Adv. pre-training [18] | 57.4 | 58.2 | 57.7 | - | 57.4[†] | 87.1 |
| Madry et al. [29] | 45.8 | - | - | 47.8 | 45.8 | 87.3 |
| Standard self-training | - | 0.3 | 0 | - | 0 | 96.4 |

Table 1: **Heuristic defense.** CIFAR-10 test accuracy under different optimization-based $\ell_\infty$ attacks of magnitude $\epsilon = 8/255$. Robust self-training (RST) with 500K unlabeled Tiny Images outperforms the state-of-the-art robust models in terms of robustness as well as standard accuracy (no attack). Standard self-training with the same data does not provide robustness. †: A projected gradient attack with 1K restarts reduces the accuracy of this model to 52.9%, evaluated on 10% of the test set [18].

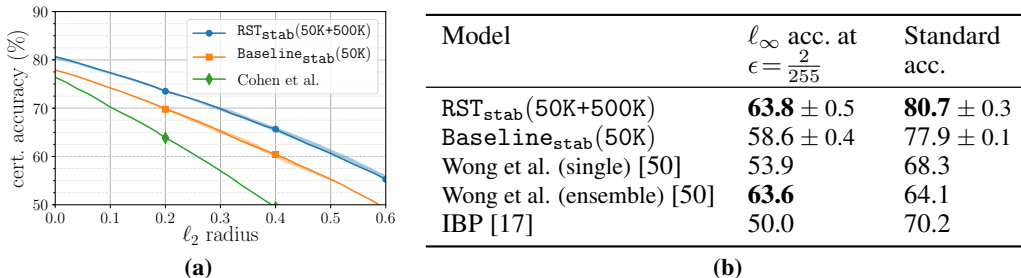

| Model | $\ell_\infty$ acc. at $\epsilon = \frac{2}{255}$ | Standard acc. |
|---|---|---|
| $\text{RST}_{\texttt{stab}}$(50K+500K) | **63.8** $\pm$ 0.5 | **80.7** $\pm$ 0.3 |
| $\texttt{Baseline}_{\texttt{stab}}$(50K) | 58.6 $\pm$ 0.4 | 77.9 $\pm$ 0.1 |
| Wong et al. (single) [50] | 53.9 | 68.3 |
| Wong et al. (ensemble) [50] | **63.6** | 64.1 |
| IBP [17] | 50.0 | 70.2 |

(a)     (b)

Figure 1: **Certified defense.** Guaranteed CIFAR-10 test accuracy under all $\ell_2$ and $\ell_\infty$ attacks. Stability-based robust self-training with 500K unlabeled Tiny Images ($\text{RST}_{\texttt{stab}}$(50K+500K)) outperforms stability training with only labeled data ($\texttt{Baseline}_{\texttt{stab}}$(50K)). **(a)** Accuracy vs. $\ell_2$ radius, certified via randomized smoothing [9]. Shaded regions indicate variation across 3 runs. Accuracy at $\ell_2$ radius 0.435 implies accuracy at $\ell_\infty$ radius 2/255. **(b)** The implied $\ell_\infty$ certified accuracy is comparable to the state-of-the-art in methods that directly target $\ell_\infty$ robustness.

**Evaluating certified defenses.** For $\text{RST}_{\texttt{stab}}$ and other models trained against random noise, we evaluate *certified* robust accuracy of the *smoothed* classifier against $\ell_2$ attacks. We perform the certification using the randomized smoothing protocol described in [9], with parameters $N_0 = 100$, $N = 10^4$, $\alpha = 10^{-3}$ and noise variance $\sigma = 0.25$.

**Evaluating variability.** We repeat training 3 times and report accuracy as X $\pm$ Y, with X the median across runs and Y half the difference between the minimum and maximum.

## 5.1 CIFAR-10

### 5.1.1 Sourcing unlabeled data

To obtain unlabeled data distributed similarly to the CIFAR-10 images, we use the 80 Million Tiny Images (80M-TI) dataset [46], of which CIFAR-10 is a manually labeled subset. However, most images in 80M-TI do not correspond to CIFAR-10 image categories. To select relevant images, we train an 11-way classifier to distinguish CIFAR-10 classes and an 11[th] "non-CIFAR-10" class using a Wide ResNet 28-10 model [54] (the same as in our experiments below). For each class, we select additional 50K images from 80M-TI using the trained model's predicted scores[3]—this is our 500K images unlabeled which we add to the 50K CIFAR-10 training set when performing RST. We provide a detailed description of the data sourcing process in Appendix B.6.

### 5.1.2 Benefit of unlabeled data

We perform robust self-training using the unlabeled data described above. We use a Wide ResNet 28-10 architecture for both the intermediate pseudo-label generator and final robust model. For adversarial training, we compute $x_{\texttt{PG}}$ exactly as in [56] with $\epsilon = 8/255$, and denote the resulting

model as $\mathtt{RST_{adv}}$(50K+500K). For stability training, we set the additive noise variance to to $\sigma = 0.25$ and denote the result $\mathtt{RST_{stab}}$(50K+500K). We provide training details in Appendix B.1.

**Robustness of $\mathtt{RST_{adv}}$(50K+500K) against strong attacks.** In Table 1, we report the accuracy of $\mathtt{RST_{adv}}$(50K+500K) and the best models in the literature against various strong attacks at $\epsilon = 8/255$ (see Appendix B.3 for details). $\mathtt{PG_{TRADES}}$ and $\mathtt{PG_{Madry}}$ correspond to the attacks used in [56] and [29] respectively, and we apply the Carlini-Wagner attack $\mathtt{CW}$ [7] on 1,000 random test examples, where we use the implementation [34] that performs search over attack hyperparameters. We also tune a PG attack against $\mathtt{RST_{adv}}$(50K+500K) (to maximally reduce its accuracy), which we denote $\mathtt{PG_{Ours}}$ (see Appendix B.3 for details).

$\mathtt{RST_{adv}}$(50K+500K) gains 7% over TRADES [56], which we can directly attribute to the unlabeled data (see Appendix B.4). In Appendix C.7 we also show this gain holds over different attack radii. The model of Hendrycks et al. [18] is based on ImageNet adversarial pretraining and is less directly comparable to ours due to the difference in external data and training method. Finally, we perform standard self-training using the unlabeled data, which offers a moderate 0.4% improvement in standard accuracy over the intermediate model but is not adversarially robust (see Appendix C.6).

**Certified robustness of $\mathtt{RST_{stab}}$(50K+500K).** Figure 1a shows the certified robust accuracy as a function of $\ell_2$ perturbation radius for different models. We compare $\mathtt{RST_{adv}}$(50K+500K) with [9], which has the highest reported certified accuracy, and $\mathtt{Baseline_{stab}}$(50K), a model that we trained using only the CIFAR-10 training set and the same training configuration as $\mathtt{RST_{stab}}$(50K+500K). $\mathtt{RST_{stab}}$(50K+500K) improves on our $\mathtt{Baseline_{stab}}$(50K) by 3–5%. The gains of $\mathtt{Baseline_{stab}}$(50K) over the previous state-of-the-art are due to a combination of better architecture, hyperparameters, and training objective (see Appendix B.5). The certified $\ell_2$ accuracy is strong enough to imply state-of-the-art certified $\ell_\infty$ robustness via elementary norm bounds. In Figure 1b we compare $\mathtt{RST_{stab}}$(50K+500K) to the state-of-the-art in certified $\ell_\infty$ robustness, showing a a 10% improvement over single models, and performance on par with the cascade approach of [50]. We also outperform the cascade model's standard accuracy by 16%.

### 5.1.3 Comparison to alternatives and ablations studies

**Consistency-based semisupervised learning (Appendix C.1).** Virtual adversarial training (VAT), a state-of-the-art method for (standard) semisupervised training of neural network [30, 33], is easily adapted to the adversarially-robust setting. We train models using adversarial- and stability-flavored adaptations of VAT, and compare them to their robust self-training counterparts. We find that the VAT approach offers only limited benefit over fully-supervised robust training, and that robust self-training offers 3–6% higher accuracy.

**Data augmentation (Appendix C.2).** In the low-data/standard accuracy regime, strong data augmentation is competitive against and complementary to semisupervised learning [10, 51], as it effectively increases the sample size by generating different plausible inputs. It is therefore natural to compare state-of-the-art data augmentation (on the labeled data only) to robust self-training. We consider two popular schemes: Cutout [13] and AutoAugment [10]. While they provide significant benefit to standard accuracy, both augmentation schemes provide essentially no improvements when we add them to our fully supervised baselines.

**Relevance of unlabeled data (Appendix C.3).** The theoretical analysis in Section 3 suggests that self-training performance may degrade significantly in the presence of irrelevant unlabeled data; other semisupervised learning methods share this sensitivity [33]. In order to measure the effect on robust self-training, we mix out unlabeled data sets with different amounts of random images from 80M-TI and compare the performance of resulting models. We find that stability training is more sensitive than adversarial training, and that both methods still yield noticeable robustness gains, even with only 50% relevant data.

**Amount of unlabeled data (Appendix C.4).** We perform robust self-training with varying amounts of unlabeled data and observe that 100K unlabeled data provide roughly half the gain provided by 500K unlabeled data, indicating diminishing returns as data amount grows. However, as we report in Appendix C.4, hyperparameter tuning issues make it difficult to assess how performance trends with data amount.

| Model | $\text{PG}_{\text{Ours}}$ | No attack |
|---|---|---|
| $\texttt{Baseline}_{\texttt{at}}(73\text{K})$ | $75.3 \pm 0.4$ | $94.7 \pm 0.2$ |
| $\texttt{RST}_{\texttt{adv}}(73\text{K}+531\text{K})$ | $86.0 \pm 0.1$ | $97.1 \pm 0.1$ |
| $\texttt{Baseline}_{\texttt{at}}(604\text{K})$ | $86.4 \pm 0.2$ | $97.5 \pm 0.1$ |

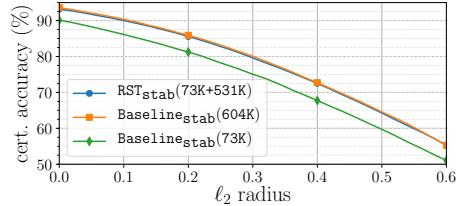

Figure 3: SVHN test accuracy for robust training without the extra data, with unlabeled extra (self-training), and with the labeled extra data. Left: Adversarial training and accuracies under $\ell_\infty$ attack with $\epsilon = 4/255$. Right: Stability training and certified $\ell_2$ accuracies as a function of perturbation radius. Most of the gains from extra data comes from the unlabeled inputs.

**Amount of labeled data (Appendix C.5).** Finally, to explore the complementary question of the effect of varying the amount of labels available for pseudo-label generation, we strip the labels of all but $n_0$ CIFAR-10 images, and combine the remainder with our 500K unlabeled data. We observe that $n_0 = 8\text{K}$ labels suffice to to exceed the robust accuracy of the (50K labels) fully-supervised baselines for both adversarial training and the $\text{PG}_{\text{Ours}}$ attack, and certified robustness via stability training.

## 5.2 Street View House Numbers (SVHN)

The SVHN dataset [53] is naturally split into a core training set of about 73K images and an 'extra' training set with about 531K easier images. In our experiments, we compare three settings: (i) robust training on the core training set only, denoted $\texttt{Baseline}_*(73\text{K})$, (ii) robust self-training with the core training set and the extra training images, denoted $\texttt{RST}_*(73\text{K}+531\text{K})$, and (iii) robust training on all the SVHN training data, denoted $\texttt{Baseline}_*(604\text{K})$. As in CIFAR-10, we experiment with both adversarial and stability training, so $*$ stands for either $\texttt{adv}$ or $\texttt{stab}$.

Beyond validating the benefit of additional data, our SVHN experiments measure the loss inherent in using pseudo-labels in lieu of true labels. Figure 3 summarizes the results: the unlabeled provides significant gains in robust accuracy, and the accuracy drop due to using pseudo-labels is below 1%. This reaffirms our intuition that in regimes of interest, *perfect labels are not crucial* for improving robustness. We give a detailed account of our SVHN experiments in Appendix D, where we also compare our results to the literature.

## 6 Related work

**Semisupervised learning.** The literature on semisupervised learning dates back to beginning of machine learning [42, 8]. A recent family of approaches operate by enforcing consistency in the model's predictions under various perturbations of the unlabeled data [30, 51], or over the course of training [45, 40, 23]. While self-training has shown some gains in standard accuracy [25], the consistency-based approaches perform significantly better on popular semisupervised learning benchmarks [33]. In contrast, our paper considers the very different regime of adversarial robustness, and we observe that robust self-training offers significant gains in robustness over fully-supervised methods. Moreover, it seems to outperform consistency-based regularization (VAT; see Section C.1). We note that there are many additional approaches to semisupervised learning, including transductive SVMs, graph-based methods, and generative modeling [8, 58].

**Self-training for domain adaptation.** Self-training is gaining prominence in the related setting of *unsupervised domain adaptation* (UDA). There, the unlabeled data is from a "target" distribution, which is different from the "source" distribution that generates labeled data. Several recent approaches [cf. 27, 19] are based on approximating class-conditional distributions of the target domain via self-training, and then learning feature transformations that match these conditional distributions across the source and target domains. Another line of work [59, 60] is based on iterative self-training coupled with refinements such as class balance or confidence regularization. Adversarial robustness and UDA share the similar goal of learning models that perform well under some kind of distribution shift; in UDA we access the target distribution through unlabeled data while in adversarial robustness, we characterize target distributions via perturbations. The fact that self-training is effective in both cases suggests it may apply to distribution shift robustness more broadly.

**Training robust classifiers.**    The discovery of adversarial examples [44, 4, 3] prompted a flurry of "defenses" and "attacks." While several defenses were broken by subsequent attacks [7, 1, 6], the general approach of adversarial training [29, 43, 56] empirically seems to offer gains in robustness. Other lines of work attain *certified* robustness, though often at a cost to empirical robustness compared to heuristics [36, 49, 37, 50, 17]. Recent work by Hendrycks et al. [18] shows that even when pre-training has limited value for standard accuracy on benchmarks, adversarial pre-training is effective. We complement this work by showing that a similar conclusion holds for semisupervised learning (both practically and theoretically in a stylized model), and extends to certified robustness as well.

**Sample complexity upper bounds.**    Recent works [52, 21, 2] study adversarial robustness from a learning-theoretic perspective, and in a number of simplified settings develop generalization bounds using extensions of Rademacher complexity. In some cases these upper bounds are demonstrably larger than their standard counterparts, suggesting there may be statistical barriers to robust learning.

**Barriers to robustness.**    Schmidt et al. [41] show a sample complexity barrier to robustness in a stylized setting. We observed that in this model, unlabeled data is as useful for robustness as labeled data. This observation led us to experiment with robust semisupervised learning. Recent work also suggests other barriers to robustness: Montasser et al. [31] show settings where improper learning and surrogate losses are crucial in addition to more samples; Bubeck et al. [5] and Degwekar and Vaikuntanathan [12] show possible computational barriers; Gilmer et al. [16] show a high-dimensional model where robustness is a consequence of any non-zero standard error, while Raghunathan et al. [38], Tsipras et al. [47], Fawzi et al. [15] show settings where robust and standard errors are at odds. Studying ways to overcome these additional theoretical barriers may translate to more progress in practice.

**Semisupervised learning for adversarial robustness.**    Independently and concurrently with our work, Zhai et al. [55], Najafi et al. [32] and Uesato et al. [48] also study the use of unlabeled data in the adversarial setting. We briefly describe each work in turn, and then contrast all three with ours.

Zhai et al. [55] study the Gaussian model of [41] and show a PCA-based procedure that successfully leverages unlabeled data to obtain adversarial robustness. They propose a training procedure that at every step treats the current model's predictions as true labels, and experiment on CIFAR-10. Their experiments include the standard semisupervised setting where some labels are removed, as well as the transductive setting where the test set is added to the training set without labels.

Najafi et al. [32] extend the distributionally robust optimization perspective of [43] to a semisupervised setting. They propose a training objective that replaces pseudo-labels with soft labels weighted according to an adversarial loss, and report results on MNIST, CIFAR-10, and SVHN with some training labels removed. The experiments in [55, 32] do not augment CIFAR-10 with new unlabeled data and do not improve the state-of-the-art in adversarial robustness.

The work of Uesato et al. [48] is the closest to ours—they also study self-training in the Gaussian model and propose a version of robust self-training which they apply on CIFAR-10 augmented with Tiny Images. Using the additional data they obtain new state-of-the-art results in heuristic defenses, comparable to ours. As our papers are very similar, we provide a detailed comparison in Appendix E.

Our paper offers a number of perspectives that complement [48, 55, 32]. First, in addition to heuristic defenses, we show gains in certified robustness where we have a guarantee on robustness against *all* possible attacks. Second, we study the impact of irrelevant unlabeled data theoretically (Section 3.3) and empirically (Appendix C.3). Finally, we provide additional experimental studies of data augmentation and of the impact of unlabeled data amount when using all labels from CIFAR-10.

## 7    Conclusion

We show that unlabeled data closes a sample complexity gap in a stylized model and that robust self-training (RST) is consistently beneficial on two image classification benchmarks. Our findings open up a number of avenues for further research. Theoretically, is sufficient unlabeled data a universal cure for sample complexity gaps between standard and adversarially robust learning? Practically, what is the best way to leverage unlabeled data for robustness, and can semisupervised learning similarly benefit alternative (non-adversarial) notions of robustness? As the scale of data grows, computational capacities increase, and machine learning moves beyond minimizing average error, we expect unlabeled data to provide continued benefit.

**Reproducibility.** Code, data, and experiments are available on GitHub at `https://github.com/yaircarmon/semisup-adv` and on CodaLab at `https://bit.ly/349WsAC`.

**Acknowledgments**

The authors would like to thank an anonymous reviewer for proposing the label amount experiment in Appendix C.5. YC was supported by the Stanford Graduate Fellowship. AR was supported by the Google Fellowship and Open Philanthropy AI Fellowship. PL was supported by the Open Philanthropy Project Award. JCD was supported by the NSF CAREER award 1553086, the Sloan Foundation and ONR-YIP N00014-19-1-2288.

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

## Footnotes

[2] Zhang et al. [56] write the regularization term $D_{\text{KL}}(p_\theta(\cdot \mid x') \,\|\, p_\theta(\cdot \mid x))$, i.e. with $p_\theta(\cdot \mid x')$ rather than $p_\theta(\cdot \mid x)$ taking role of the label, but their open source implementation follows (6).

[3]We exclude any image close to the CIFAR-10 test set; see Appendix B.6 for detail.
