[Supplementary Material]

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

 $\texttt{RST}_{\texttt{stab}}(50\text{K}+500\text{K})$. $\texttt{RST}_{\texttt{stab}}(50\text{K}+500\text{K})$ improves on our $\texttt{Baseline}_{\texttt{stab}}(50\text{K})$ by 3–5%. The gains of $\texttt{Baseline}_{\texttt{stab}}(50\text{K})$ over the previous state-of-the-art are due to a combination of better architecture, hyperparameters, and training objective (see Appendix B.5). The certified $\ell_2$ accuracy is strong enough to imply state-of-the-art certified $\ell_\infty$ robustness via elementary norm bounds. In Figure 1b we compare $\texttt{RST}_{\texttt{stab}}(50\text{K}+500\text{K})$ to the state-of-the-art in certified $\ell_\infty$ robustness, showing a a 10% improvement over single models, and performance on par with the cascade approach of [50]. We also outperform the cascade model's standard accuracy by $16\%$.

### 5.1.3 Comparison to alternatives and ablations studies

**Consistency-based semisupervised learning (Appendix C.1).** Virtual adversarial training (VAT), a state-of-the-art method for (standard) semisupervised training of neural network [30, 33], is easily adapted to the adversarially-robust setting. We train models using adversarial- and stability-flavored adaptations of VAT, and compare them to their robust self-training counterparts. We find that the VAT approach offers only limited benefit over fully-supervised robust training, and that robust self-training offers 3–6% higher accuracy.

**Data augmentation (Appendix C.2).** In the low-data/standard accuracy regime, strong data augmentation is competitive against and complementary to semisupervised learning [10, 51], as it effectively increases the sample size by generating different plausible inputs. It is therefore natural to compare state-of-the-art data augmentation (on the labeled data only) to robust self-training. We consider two popular schemes: Cutout [13] and AutoAugment [10]. While they provide significant benefit to standard accuracy, both augmentation schemes provide essentially no improvements when we add them to our fully supervised baselines.

**Relevance of unlabeled data (Appendix C.3).** The theoretical analysis in Section 3 suggests that self-training performance may degrade significantly in the presence of irrelevant unlabeled data; other semisupervised learning methods share this sensitivity [33]. In order to measure the effect on robust self-training, we mix out unlabeled data sets with different amounts of random images from 80M-TI and compare the performance of resulting models. We find that stability training is more sensitive than adversarial training, and that both methods still yield noticeable robustness gains, even with only 50% relevant data.

**Amount of unlabeled data (Appendix C.4).** We perform robust self-training with varying amounts of unlabeled data and observe that 100K unlabeled data provide roughly half the gain provided by 500K unlabeled data, indicating diminishing returns as data amount grows. However, as we report in Appendix C.4, hyperparameter tuning issues make it difficult to assess how performance trends with data amount.

| Model | $PG_{0urs}$ | No attack |
|---|---|---|
| Baseline$_{at}$(73K) | $75.3 \pm 0.4$ | $94.7 \pm 0.2$ |
| RST$_{adv}$(73K+531K) | $86.0 \pm 0.1$ | $97.1 \pm 0.1$ |
| Baseline$_{at}$(604K) | $86.4 \pm 0.2$ | $97.5 \pm 0.1$ |

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

# Supplementary Material

## A   Theoretical results

This appendix contains the full proofs for the results in Section 3, as well as explicit bounds for the robust error of the self-training estimator.

We remark that the results of this section easily extend to the case where there is class imbalance: The upper bounds in Proposition 1 and Theorem 2 hold regardless of the label distribution, while the lower bound in Theorem 1 changes from $\frac{1}{2}(1-d^{-1})$ to $p(1-d^{-1})$ where $p$ is the proportion of the smaller class; the only change to the proof in [41] is a modification of the lower bound on $\Psi$ in page 29 of the arxiv version.

### A.1   Error probabilities in closed form

We recall our model $x \sim \mathcal{N}(y\mu, \sigma^2 I)$ with $y$ uniform on $\{-1, 1\}$ and $\mu \in \mathbb{R}^n$. Consider a linear classifier $f_\theta(x) = \text{sign}(x^\top \theta)$. Then the standard error probability is

$$\text{err}_{\text{standard}}(f_\theta) = \mathbb{P}(y \cdot x^\top \theta < 0) = \mathbb{P}\left(\mathcal{N}\left(\frac{\mu^\top \theta}{\sigma\|\theta\|}, 1\right) < 0\right) =: Q\left(\frac{\mu^\top \theta}{\sigma\|\theta\|}\right) \tag{10}$$

where

$$Q(x) = \frac{1}{\sqrt{2\pi}} \int_x^\infty e^{-t^2/2} dt$$

is the Gaussian error function. For linear classifier $f_\theta$, input $x$ and label $y$, the strongest adversarial perturbation of $x$ with $\ell_\infty$ norm $\epsilon$ moves each coordinate of $x$ by $-\epsilon \,\text{sign}(y\theta)$. The robust error probability is therefore

$$\text{err}_{\text{robust}}^{\infty,\epsilon}(f_\theta) = \mathbb{P}\left(\inf_{\|\nu\|_\infty \leq \epsilon} \left\{y \cdot (x+\nu)^\top \theta\right\} < 0\right)$$

$$= \mathbb{P}(y \cdot x^\top \theta - \epsilon\|\theta\|_1 < 0) = \mathbb{P}(\mathcal{N}(\mu^\top\theta, (\sigma\|\theta\|)^2) < \epsilon\|\theta\|_1)$$

$$= Q\left(\frac{\mu^\top\theta}{\sigma\|\theta\|} - \frac{\epsilon\|\theta\|_1}{\sigma\|\theta\|}\right) \leq Q\left(\frac{\mu^\top\theta}{\sigma\|\theta\|} - \frac{\epsilon\sqrt{d}}{\sigma}\right). \tag{11}$$

In this model, standard and robust accuracies align in the sense that any highly accurate standard classifier, with $\frac{\mu^\top\theta}{\|\theta\|} > \epsilon\sqrt{d}$, will necessarily also be robust. Moreover, for dense $\mu$ (with $\|\mu\|_1/\|\mu\| = \Omega(\sqrt{d})$), good linear estimators will typically be dense as well, in which case $\frac{\mu^\top\theta}{\sigma\|\theta\|}$ determines both standard and robust accuracies. Our analysis will consequently focus on understanding the quantity $\frac{\mu^\top\theta}{\sigma\|\theta\|}$.

#### A.1.1   Optimal standard accuracy and parameter setting

We note that for a given problem instance, the classifier that minimizes the standard error is simply $\theta^\star = \mu$. Its standard error is

$$\text{err}_{\text{standard}}(f_{\theta^\star}) = Q\left(\frac{\|\mu\|}{\sigma}\right) \leq e^{-\|\mu\|^2/2\sigma^2}.$$

Recall our parameter setting,

$$\epsilon \leq \frac{1}{2}, \ \sigma = (n_0 d)^{1/4}, \text{ and } \|\mu\|^2 = d. \tag{12}$$

Under this setting, $\frac{\|\mu\|}{\sigma} = \left(\frac{d}{n_0}\right)^{1/4}$ and we have

$$\text{err}_{\text{standard}}(f_{\theta^\star}) = Q\left(\left(\frac{d}{n_0}\right)^{1/4}\right) \leq e^{-\frac{1}{2}\sqrt{d/n_0}} \text{ and } \text{err}_{\text{robust}}^{\infty,\epsilon}(f_{\theta^\star}) \leq Q\left((1-\epsilon)\left(\frac{d}{n_0}\right)^{1/4}\right) \leq e^{-\frac{1}{8}\sqrt{d/n_0}}.$$

Therefore, in the regime $d/n_0 \gg 1$, the classifier $\theta^\star$ achieves essentially perfect accuracies, both standard and robust. We will show that estimating $\theta$ from $n_0$ labeled data and a large number ($\approx \sqrt{d/n_0}$) of unlabeled data allows us to approach the performance of $\theta^\star$, without prior knowledge of $\mu$.

## A.2 Performance of supervised estimator

Given labeled data set $(x_1,y_1),...,(x_n,y_n)$ we consider the linear classifier given by

$$\hat{\theta}_n = \frac{1}{n}\sum_{i=1}^{n} y_i x_i.$$

In the following lemma we give a tight concentration bound for $\mu^\top\hat{\theta}_n / \left(\sigma\left\|\hat{\theta}_n\right\|\right)$, which determines the standard and robust error probabilities of $f_{\hat{\theta}_n}$ via equations (10) and (11) respectively

**Lemma 1.** *There exist numerical constants $c_0,c_1,c_2$ such that under parameter setting (12) and $d/n_0 > c_0$,*

$$\frac{\mu^\top\hat{\theta}_n}{\sigma\left\|\hat{\theta}_n\right\|} \geq \left(\sqrt{\frac{n_0}{d}} + \frac{n_0}{n}\left(1+c_1\left(\frac{n_0}{d}\right)^{1/8}\right)\right)^{-1/2} \text{ with probability } \geq 1 - e^{-c_2(d/n_0)^{1/4}\min\{n,(d/n_0)^{1/4}\}}.$$

*Proof.* We have

$$\hat{\theta}_n \sim \mathcal{N}\left(\mu, \frac{\sigma^2}{n}I\right) \text{ so that } \delta := \hat{\theta}_n - \mu \sim \mathcal{N}\left(0, \frac{\sigma^2}{n}I\right).$$

To lower bound the random variable $\frac{\mu^\top\hat{\theta}_n}{\|\hat{\theta}_n\|}$ we consider its squared inverse, and decompose it as follows

$$\frac{\left\|\hat{\theta}_n\right\|^2}{\left(\mu^\top\hat{\theta}_n\right)^2} = \frac{\|\delta+\mu\|^2}{\left(\|\mu\|^2+\mu^\top\delta\right)^2} = \frac{1}{\|\mu\|^2} + \frac{\|\delta\|^2 - \frac{1}{\|\mu\|^2}\left(\mu^\top\delta\right)^2}{\left(\|\mu\|^2+\mu^\top\delta\right)^2}$$

$$\leq \frac{1}{\|\mu\|^2} + \frac{\|\delta\|^2}{\left(\|\mu\|^2+\mu^\top\delta\right)^2}$$

To obtain concentration bounds, we note that

$$\|\delta\|^2 \sim \frac{\sigma^2}{n}\chi_d^2 \text{ and } \frac{\mu^\top\delta}{\|\mu\|} \sim \mathcal{N}\left(0, \frac{\sigma^2}{n}\right).$$

Therefore, standard concentration results give

$$\mathbb{P}\left(\|\delta\|^2 \geq \frac{\sigma^2}{n}\left(d+\frac{1}{\sigma}\right)\right) \leq e^{-d/8\sigma^2} \text{ and } \mathbb{P}\left(\frac{\mu^\top\delta}{\|\mu\|} \geq (\sigma\|\mu\|)^{1/2}\right) \leq 2e^{-\frac{1}{2}n\|\mu\|/\sigma}. \quad (13)$$

Assuming that the two events $\|\delta\|^2 \leq \frac{\sigma^2}{n}\left(d+\frac{1}{\sigma}\right)$ and $\left|\mu^\top\delta\right| \leq \sigma^{1/2}\|\mu\|^{3/2}$ hold, we have

$$\frac{\left\|\hat{\theta}_n\right\|^2}{\left(\mu^\top\hat{\theta}_n\right)^2} \leq \frac{1}{\|\mu\|^2} + \frac{\frac{\sigma^2}{n}\left(d+\frac{1}{\sigma}\right)}{\|\mu\|^4\left(1-(\sigma/\|\mu\|)^{-1/2}\right)^2}.$$

Substituting the parameter setting setting (12), we have that for $d/n_0$ sufficiently large,

$$\frac{\sigma^2\left\|\hat{\theta}_n\right\|^2}{\left(\mu^\top\hat{\theta}_n\right)^2} \leq \sqrt{\frac{n_0}{d}} + \frac{\frac{n_0 d}{n}\left(d+(n_0 d)^{-1/4}\right)}{d^2\left(1-(n_0/d)^{1/8}\right)^2} \leq \sqrt{\frac{n_0}{d}} + \frac{n_0}{n}\left(1+c_1(n_0/d)^{1/8}\right)$$

for some numerical constant $c_1$. For this to imply the bound stated in the lemma we also need $\mu^\top\hat{\theta}_n \geq 0$ to hold, but this is already implied by

$$\mu^\top\hat{\theta}_n = \|\mu\|^2 + \mu^\top\delta \geq \|\mu\|^2\left(1-(\sigma/\|\mu\|)^{-1/2}\right) \geq d\left(1-\left(\frac{n_0}{d}\right)^{1/8}\right) > 0.$$

Substituting the parameters settings into the concentration bounds (13), we have by the union bound that the desired upper bound fails to hold with probability at most

$$e^{-d/8\sqrt{n_0 d}} + 2e^{-n\sqrt{d}/2(n_0 d)^{1/4}} \leq e^{-c_2(d/n_0)^{1/4}\min\{n,(d/n_0)^{1/4}\}}$$

for another numerical constant $c_2$ and $d/n_0 > 1$. $\qquad\square$

As an immediate corollary to Lemma 1, we obtain the sample complexity upper bounds cited in the main text.

**Proposition 1.** *There exists a universal constant $r$ such that for all $\epsilon^2\sqrt{d/n_0}\geq r$,*

$$n\geq n_0 \Rightarrow \mathbb{E}_{\hat{\theta}_n}\,\mathrm{err}_{\mathrm{standard}}\left(f_{\hat{\theta}_n}\right)\leq\frac{1}{3} \ \ and \ \ n\geq n_0\cdot 4\epsilon^2\sqrt{\frac{d}{n_0}} \Rightarrow \mathbb{E}_{\hat{\theta}_n}\,\mathrm{err}_{\mathrm{robust}}^{\infty,\epsilon}\left(f_{\hat{\theta}_n}\right)\leq 10^{-3}.$$

*Proof.* For the case $n\geq n_0$ we take $r$ sufficiently large such that by Lemma 1 we have

$$\frac{\mu^\top\hat{\theta}_n}{\sigma\left\|\hat{\theta}_n\right\|}\geq\frac{1}{\sqrt{2\left(\frac{n_0}{n}+\sqrt{\frac{n_0}{d}}\right)}}\geq\frac{1}{2} \text{ with probability}\geq 1-e^{-c_2\sqrt{d/n_0}}$$

for an appropriate $c_2$. Therefore by the expression (10) for the standard error probability (and the fact that it is never more than 1), we have

$$\mathbb{E}_{\hat{\theta}_n}\,\mathrm{err}_{\mathrm{standard}}\left(f_{\hat{\theta}_n}\right)\leq Q\left(\frac{1}{2}\right)+e^{-c_2(d/n_0)^{1/8}}\leq\frac{1}{3}$$

for appropriate $r$. Similarly, for the case $n\geq n_0\cdot 4\epsilon^2\sqrt{\frac{d}{n_0}}$ we apply Lemma 1 combined with $\epsilon<\frac{1}{2}$ to write

$$\frac{\mu^\top\hat{\theta}_n}{\sigma\left\|\hat{\theta}_n\right\|}\geq\frac{1}{\sqrt{2\left(\frac{n_0}{n}+\sqrt{\frac{n_0}{d}}\right)}}\geq\frac{1}{\sqrt{2\left(\frac{n_0}{4\epsilon^2\sqrt{n_0 d}}+\frac{1}{4\epsilon^2}\sqrt{\frac{n_0}{d}}\right)}}=\sqrt{2}\epsilon\left(\frac{d}{n_0}\right)^{1/4}$$

with probability $\geq 1-e^{-c_2(d/n_0)^{1/4}\min\{n,(d/n_0)^{1/4}\}}$. Therefore, using the expression (11) and $\sigma=(n_0 d)^{1/4}$, we have (using $n\geq\epsilon^2(d/n_0)^{1/4}$)

$$\mathbb{E}_{\hat{\theta}_n}\,\mathrm{err}_{\mathrm{robust}}^{\infty,\epsilon}\left(f_{\hat{\theta}_n}\right)\leq Q\left(\left[\sqrt{2}-1\right]\epsilon(d/n_0)^{1/4}\right)+e^{-\epsilon^2 c_2\sqrt{d/n_0}}\leq 10^{-3},$$

for sufficiently large $r$. $\qquad\square$

### A.3 Lower bound

We now briefly explain how to translate the sample complexity lower bound of Schmidt et al. [41] into our parameter setting.

**Theorem 1** ([41]). *Let $\mathsf{A}_n$ be any learning rule mapping a dataset $S\in(\mathcal{X}\times\mathcal{Y})^n$ to classifier $\mathsf{A}_n[S]$. Then,*

$$n\leq n_0\frac{\epsilon^2\sqrt{d/n_0}}{8\log d} \Rightarrow \mathbb{E}\mathrm{err}_{\mathrm{robust}}^{\infty,\epsilon}(\mathsf{A}_n[S])\geq\frac{1}{2}(1-d^{-1}), \tag{5}$$

*where the expectation is with respect to the random draw of $S\sim P_{\mathsf{x},\mathsf{y}}^n$ as well as possible randomization in $\mathsf{A}_n$.*

*Proof.* The setting of our theorem is identical to that of Theorem 11 in Schmidt et al. [41], which shows that

$$\mathbb{E}\mathrm{err}_{\mathrm{robust}}^{\infty,\epsilon}(\mathsf{A}_n[S])\geq\frac{1}{2}\mathbb{P}\left(\|\mathcal{N}(0,I)\|_\infty\leq\epsilon\sqrt{1+\frac{\sigma^2}{n}}\right).$$

Using $\sigma^2=\sqrt{n_0 d}$, $n\leq\frac{\epsilon^2\sqrt{n_0 d}}{8\log d}$ implies $\epsilon\sqrt{1+\frac{\sigma^2}{n}}\geq\sqrt{8\log d}$ and therefore

$$\mathbb{E}\mathrm{err}_{\mathrm{robust}}^{\infty,\epsilon}(\mathsf{A}_n[S])\geq\frac{1}{2}\mathbb{P}\left(\|\mathcal{N}(0,I)\|_\infty\leq\sqrt{8\log d}\right).$$

Moreover

$$\mathbb{P}\left(\|\mathcal{N}(0,I)\|_\infty\leq\sqrt{8\log d}\right)=\left(1-Q\left(\sqrt{8\log d}\right)\right)^d\geq\left(1-e^{-4\log d}\right)^d\geq 1-\frac{1}{d}.$$

$\qquad\square$

## A.4 Performance of semisupervised estimator

We now consider the semisupervised setting—our primary object of study in this paper. We consider the self-training estimator that in the first stage uses $n \geq n_0$ labeled examples to construct

$$\hat{\theta}_{\text{intermediate}} := \hat{\theta}_n,$$

and then uses it to produce pseudo-labels

$$\tilde{y}_i = \text{sign}\left(\tilde{x}_i^\top \hat{\theta}_{\text{intermediate}}\right)$$

for the $\tilde{n}$ unlabeled data points $\tilde{x}_1, \ldots, \tilde{x}_{\tilde{n}}$. In the second and final stage of self-training, we employ the same simple learning rule on the pseudo-labeled data and construct

$$\hat{\theta}_{\text{final}} := \frac{1}{\tilde{n}} \sum_{i=1}^{\tilde{n}} \tilde{y}_i \tilde{x}_i.$$

The following result shows a high-probability bound on $\frac{\mu^\top \hat{\theta}_{\text{final}}}{\sigma \|\hat{\theta}_{\text{final}}\|}$, analogous to the one obtained for the fully supervised estimator in Lemma 1 (with different constant factors).

**Lemma 2.** *There exist numerical constants $\tilde{c}_0, \tilde{c}_1, \tilde{c}_2 > 0$ such that under parameter setting (12) and $d/n_0 > \tilde{c}_0$,*

$$\frac{\mu^\top \hat{\theta}_{\text{final}}}{\sigma \|\hat{\theta}_{\text{final}}\|} \geq \left( \sqrt{\frac{n_0}{d}} + \frac{72 n_0}{\tilde{n}} \left( 1 + \tilde{c}_1 \left( \frac{n_0}{d} \right)^{-1/4} \right) \right)^{-1/2}$$

*with probability $\geq 1 - e^{-\tilde{c}_2 \min\left\{ \tilde{n}, n_0 (d/n_0)^{1/4}, \sqrt{d/n_0} \right\}}$.*

*Proof.* The proof follows a similar argument to the one used to prove Lemma 1, except now we have to to take care of the fact that the noise component in $\hat{\theta}_{\text{final}}$ is not entirely Gaussian. Let $b_i$ be the indicator that the $i$th pseudo-label is incorrect, so that $\tilde{x}_i \sim \mathcal{N}\left( (1 - 2b_i) \tilde{y}_i \mu, \sigma^2 I \right)$, and let

$$\gamma := \frac{1}{\tilde{n}} \sum_{i=1}^{\tilde{n}} (1 - 2b_i) \in [-1, 1].$$

We may write the final estimator as

$$\hat{\theta}_{\text{final}} = \frac{1}{\tilde{n}} \sum_{i=1}^{\tilde{n}} \tilde{y}_i \tilde{x}_i = \gamma \mu + \frac{1}{\tilde{n}} \sum_{i=1}^{\tilde{n}} \tilde{y}_i \varepsilon_i$$

where $\varepsilon_i \sim \mathcal{N}(0, \sigma^2 I)$ independent of each other. Defining

$$\tilde{\delta} := \hat{\theta}_{\text{final}} - \gamma \mu$$

we have the decomposition and bound

$$\frac{\left\| \hat{\theta}_{\text{final}} \right\|^2}{\left( \mu^\top \hat{\theta}_{\text{final}} \right)^2} = \frac{\left\| \tilde{\delta} + \gamma \mu \right\|^2}{\left( \gamma \|\mu\|^2 + \mu^\top \tilde{\delta} \right)^2} = \frac{1}{\|\mu\|^2} + \frac{\left\| \tilde{\delta} + \gamma \mu \right\|^2 - \frac{1}{\|\mu\|^2} \left( \gamma \|\mu\|^2 + \mu^\top \tilde{\delta} \right)^2}{\left( \gamma \|\mu\|^2 + \mu^\top \tilde{\delta} \right)^2}$$

$$= \frac{1}{\|\mu\|^2} + \frac{\|\tilde{\delta}\|^2 - \frac{1}{\|\mu\|^2} \left( \mu^\top \tilde{\delta} \right)^2}{\left( \gamma \|\mu\|^2 + \mu^\top \tilde{\delta} \right)} \leq \frac{1}{\|\mu\|^2} + \frac{\|\tilde{\delta}\|^2}{\|\mu\|^4 \left( \gamma + \frac{1}{\|\mu\|^2} \mu^\top \tilde{\delta} \right)^2}. \qquad (14)$$

To write down concentration bounds for $\|\tilde{\delta}\|^2$ and $\mu^\top \tilde{\delta}$ we must address their non-Gaussianity. To do so, choose a coordinate system such that the first coordinate is in the direction of $\hat{\theta}_{\text{intermediate}}$, and let $v^{(i)}$ denote the $i$th entry of vector $v$ in this coordinate system. Then

$$\tilde{y}_i = \text{sign}\left( \tilde{x}_i^{(1)} \right) = \text{sign}\left( \mu^{(1)} + \varepsilon_i^{(1)} \right).$$

Consequently, $\varepsilon_i^{(j)}$ is independent of $\tilde{y}_i$ for all $i$ and $j \geq 2$, so that $\tilde{y}_i \varepsilon_i^{(j)} \sim \mathcal{N}\left(0, \sigma^2\right)$ and $\frac{1}{\tilde{n}} \sum_{i=1}^{\tilde{n}} \tilde{y}_i \varepsilon_i^{(j)} \sim \mathcal{N}\left(0, \sigma^2/\tilde{n}\right)$ and

$$\sum_{j=2}^{d} \left( \frac{1}{\tilde{n}} \sum_{i=1}^{\tilde{n}} \tilde{y}_i \varepsilon_i^{(j)} \right)^2 \sim \frac{\sigma^2}{\tilde{n}} \chi_{d-1}^2.$$

Moreover, we have by Cauchy–Schwarz

$$\left( \frac{1}{\tilde{n}} \sum_{i=1}^{\tilde{n}} \tilde{y}_i \varepsilon_i^{(1)} \right)^2 \leq \frac{1}{\tilde{n}^2} \left( \sum_{i=1}^{\tilde{n}} \tilde{y}_i^2 \right) \left( \sum_{i=1}^{\tilde{n}} \left[\varepsilon_i^{(1)}\right]^2 \right) = \frac{1}{\tilde{n}} \sum_{i=1}^{\tilde{n}} \left[\varepsilon_i^{(1)}\right]^2 \sim \frac{\sigma^2}{\tilde{n}} \chi_{\tilde{n}}^2.$$

Therefore, since $\|\tilde{\delta}\|^2 = \sum_{j=1}^{d} \left( \frac{1}{\tilde{n}} \sum_{i=1}^{\tilde{n}} \tilde{y}_i \varepsilon_i^{(j)} \right)^2$, we have by the union bound

$$\mathbb{P}\left( \|\tilde{\delta}\|^2 \geq 2\frac{\sigma^2}{\tilde{n}}(d-1+\tilde{n}) \right) \leq \mathbb{P}\left( \chi_{\tilde{n}}^2 \geq 2\tilde{n} \right) + \mathbb{P}\left( \chi_{d-1}^2 \geq 2(d-1) \right) \leq e^{-\tilde{n}/8} + e^{-(d-1)/8}. \tag{15}$$

The same technique also yields a crude bound on $\mu^\top \tilde{\delta} = \frac{1}{\tilde{n}} \sum_{i=1}^{\tilde{n}} \tilde{y}_i \mu^\top \varepsilon_i$. Namely, we have

$$\left( \mu^\top \tilde{\delta} \right)^2 \leq \frac{1}{\tilde{n}^2} \left( \sum_{i=1}^{\tilde{n}} \tilde{y}_i^2 \right) \left( \sum_{i=1}^{\tilde{n}} (\mu^\top \varepsilon_i)^2 \right) = \frac{1}{\tilde{n}} \sum_{i=1}^{\tilde{n}} (\mu^\top \varepsilon_i)^2 \sim \frac{\sigma^2 \|\mu\|^2}{\tilde{n}} \chi_{\tilde{n}}^2$$

and therefore

$$\mathbb{P}\left( \left| \mu^\top \tilde{\delta} \right| \geq \sqrt{2}\sigma\|\mu\| \right) = \mathbb{P}\left( \left| \mu^\top \tilde{\delta} \right|^2 \geq 2\sigma^2\|\mu\|^2 \right) \leq e^{-\tilde{n}/8}.$$

Finally, we need to argue that $\gamma$ is not too small. Recall that $\gamma = \frac{1}{\tilde{n}} \sum_{i=1}^{\tilde{n}} (1-2b_i)$ where $b_i$ is the indicator that $\tilde{y}_i$ is incorrect and therefore

$$\mathbb{E}\left[ \gamma \mid \hat{\theta}_{\text{intermediate}} \right] = 1 - 2\,\text{err}_{\text{standard}}(f_{\hat{\theta}_{\text{intermediate}}}),$$

so we expect $\gamma$ to be reasonably large as long as $\text{err}_{\text{standard}}(f_{\hat{\theta}_{\text{intermediate}}}) < \frac{1}{2}$. Indeed,

$$\mathbb{P}\left( \gamma < \frac{1}{6} \right) = \mathbb{P}\left( \frac{1}{\tilde{n}} \sum_{i=1}^{\tilde{n}} (1-2b_i) < \frac{1}{6} \right)$$

$$\leq \mathbb{P}\left( \text{err}_{\text{standard}}(f_{\hat{\theta}_{\text{intermediate}}}) > \frac{1}{3} \right) + \mathbb{P}\left( \frac{1}{\tilde{n}} \sum_{i=1}^{\tilde{n}} b_i < \frac{5}{12} \mid \text{err}_{\text{standard}}(f_{\hat{\theta}_{\text{intermediate}}}) \leq \frac{1}{3} \right).$$

Note that

$$\frac{1}{3} \geq Q\left( \frac{1}{2} \right) \geq Q\left( \left[ 2\left(1+\sqrt{n_0/d}\right) \right]^{-1/2} \right)$$

Therefore, by Lemma 1, for sufficiently large $d/n_0$,

$$\mathbb{P}\left( \text{err}_{\text{standard}}(f_{\hat{\theta}_{\text{intermediate}}}) > \frac{1}{3} \right) \leq e^{-c \cdot \min\left\{ \sqrt{d/n_0}, n_0(d/n_0)^{1/4} \right\}}$$

for some constant $c$. Moreover, by Bernoulli concentration (Hoeffding's inequality) we have that

$$\mathbb{P}\left( \frac{1}{\tilde{n}} \sum_{i=1}^{\tilde{n}} b_i < \frac{5}{12} \mid \text{err}_{\text{standard}}(f_{\hat{\theta}_{\text{intermediate}}}) \leq \frac{1}{3} \right) \leq e^{-2\tilde{n}\left( \frac{5}{12} - \frac{1}{3} \right)^2} = e^{-\tilde{n}/72}.$$

Define the event,

$$\mathcal{E} = \left\{ \|\tilde{\delta}\|^2 \geq 2\frac{\sigma^2}{\tilde{n}}(d+\tilde{n}), \left| \mu^\top \tilde{\delta} \right| \leq \sqrt{2}\sigma\|\mu\| \text{ and } \gamma \geq \frac{1}{6} \right\};$$

by the preceding discussion,

$$\mathbb{P}\left( \mathcal{E}^C \right) \leq 2e^{-\tilde{n}/8} + e^{-(d-1)/8} + e^{-c \cdot \min\left\{ \sqrt{d/n_0}, n_0(d/n_0)^{1/4} \right\}} + e^{-\tilde{n}/72} \leq e^{-\tilde{c}_2 \min\left\{ \tilde{n}, \sqrt{d/n_0}, n_0(d/n_0)^{1/4} \right\}}.$$

Moreover, by the bound (14), $\mathcal{E}$ implies

$$\frac{\left\|\hat{\theta}_{\text{final}}\right\|^2}{\left(\mu^\top \hat{\theta}_{\text{final}}\right)^2} \leq \frac{1}{\|\mu\|^2} + \frac{2\sigma^2(d+\tilde{n})}{\tilde{n}\|\mu\|^4 \left(\frac{1}{6} - \frac{\sqrt{2}\sigma}{\|\mu\|}\right)^2}.$$

Substituting $\sigma = (n_0 d)^{1/4}$ and $\|\mu\| = \sqrt{d}$ and multiplying by $\sigma^2$ gives

$$\frac{\sigma^2 \left\|\hat{\theta}_{\text{final}}\right\|^2}{\left(\mu^\top \hat{\theta}_{\text{final}}\right)^2} \leq \sqrt{\frac{n_0}{d}} + \frac{2(n_0 d)(d+\tilde{n})}{\tilde{n}d^2 \left(\frac{1}{6} - \sqrt{2}\left(\frac{n_0}{d}\right)^{1/4}\right)^2}$$

$$\leq \sqrt{\frac{n_0}{d}} + \frac{72 n_0}{\tilde{n}}\left(1 + \tilde{c}_1\left(\frac{n_0}{d}\right)^{-1/4}\right)$$

for appropriate $\tilde{c}_1$ and sufficiently large $d/n_0$. As argued in Lemma 1, the event $\mathcal{E}$ already implies $\mu^\top \hat{\theta}_{\text{final}} \geq 0$, and therefore the result follows. $\qquad\square$

Lemma 2 immediately gives a sample complexity upper bound for the self-training classifier $\hat{\theta}_{\text{final}}$ trained with $n$ labeled data and $\tilde{n}$ unlabeled data.

**Theorem 2.** *There exists a universal constant $\tilde{r}$ such that for $\epsilon^2 \sqrt{d/n_0} \geq \tilde{r}$, $n \geq n_0$ labeled data and additional $\tilde{n}$ unlabeled data,*

$$\tilde{n} \geq n_0 \cdot 288\epsilon^2 \sqrt{\frac{d}{n_0}} \Rightarrow \mathbb{E}_{\hat{\theta}_{\text{final}}} \text{err}_{\text{robust}}^{\infty,\epsilon}\left(f_{\hat{\theta}_{\text{final}}}\right) \leq 10^{-3}.$$

*Proof.* We take $\tilde{r}$ sufficiently large so that by Lemma 2 we have, using $\sigma = (n_0 d)^{1/4}$ and $\epsilon < \frac{1}{2}$,

$$\frac{\mu^\top \hat{\theta}_n}{\sigma \left\|\hat{\theta}_n\right\|} \geq \frac{1}{\sqrt{2\left(\frac{72 n_0}{n} + \sqrt{\frac{n_0}{d}}\right)}} \geq \frac{1}{\sqrt{2\left(\frac{n_0}{4\epsilon^2\sqrt{n_0 d}} + \frac{1}{4\epsilon^2}\sqrt{\frac{n_0}{d}}\right)}} = \sqrt{2}\epsilon\left(\frac{d}{n_0}\right)^{1/4}$$

with probability $\geq 1 - e^{-\tilde{c}_2 \min\left\{\tilde{n}, n_0(d/n_0)^{1/4}, \sqrt{d/n_0}\right\}} \geq 1 - e^{-\epsilon^2 \tilde{c}_2 (d/n_0)^{1/4}}$. Therefore, using the expression (11) and $\sigma = (n_0 d)^{1/4}$, we have (using $n \geq \epsilon^2(d/n_0)^{1/4}$)

$$\mathbb{E}_{\hat{\theta}_n} \text{err}_{\text{robust}}^{\infty,\epsilon}\left(f_{\hat{\theta}_n}\right) \leq Q\left(\left[\sqrt{2}-1\right]\epsilon(d/n_0)^{1/4}\right) + e^{-\epsilon^2 \tilde{c}_2 (d/n_0)^{1/4}} \leq 10^{-3},$$

for sufficiently large $\tilde{r}$. $\qquad\square$

### A.5 Performance in the presence of irrelevant data

To model the presence of irrelevant data, we consider a slightly different model where, for $\alpha \in (0,1)$, $\alpha\tilde{n}$ of the unlabeled data are distributed as $\mathcal{N}(y_i\mu, \sigma^2 I)$ as before, while the other $(1-\alpha)\tilde{n}$ unlabeled data are drawn from $\mathcal{N}(0, \sigma^2 I)$ (with no signal component). We note that similar conclusions would hold if we let the irrelevant unlabeled data be drawn from $\mathcal{N}(\mu_2, \sigma^2 I)$ for some $\mu_2$ such that $|\mu^\top \mu_2|$ is sufficiently small, for example $\mu_2 \sim \mathcal{N}(0, I)$ independent of $\mu$. We take $\mu_2 = 0$ to simplify the presentation.

To understand the impact of irrelevant data we need to establish two statements. First, we would like to show that adversarial robustness is still possible given sufficiently large $\tilde{n}$, namely $\Omega(\epsilon^2\sqrt{n_0 d}/\alpha^2)$; a factor $1/\alpha$ more relevant data then what our previous result required. Second, we wish to show that this upper bound is tight. That is, we would like to show that self-training with $n_0$ labeled data and $O(\epsilon^2\sqrt{n_0 d}/\alpha^2)$ $\alpha$-relevant unlabeled data fails to achieve robustness. We make these statements rigorous in the following.

**Theorem 3.** *There exist numerical constants $c$ and $r$ such the following holds under parameter setting (12), $\alpha$-fraction of relevant unlabeled data and $\min\{\epsilon^2/\log d, \alpha^2\}\sqrt{d/n_0} > r$. First,*

$$\tilde{n} \geq n_0 \cdot \frac{288\epsilon^2}{\alpha^2}\sqrt{\frac{d}{n_0}} \Rightarrow \mathbb{E}_{\hat{\theta}_{\text{final}}} \text{err}_{\text{robust}}^{\infty,\epsilon}\left(f_{\hat{\theta}_{\text{final}}}\right) \leq 10^{-3}.$$

*Second, there exists $\mu \in \mathbb{R}^d$ for which*

$$\tilde{n} \leq n_0 \frac{c \cdot \epsilon^2}{\alpha^2} \sqrt{\frac{d}{n_0}} \Rightarrow \mathbb{E}_{\hat{\theta}_{\text{final}}} \text{err}_{\text{robust}}^{\infty,\epsilon}\left(f_{\hat{\theta}_{\text{final}}}\right) \geq \frac{1}{2}\left(1 - \frac{1}{d}\right).$$

Examining the robust error probability (11), establishing these results requires upper and lower bounds on the quantity $\frac{\mu^\top \hat{\theta}_{\text{final}}}{\|\hat{\theta}_{\text{final}}\|}$ as well as a lower bound on $\frac{\|\hat{\theta}_{\text{final}}\|_1}{\|\hat{\theta}_{\text{final}}\|}$. We begin with the former, which is a two-sided version of Lemma 2.

**Lemma 3.** *There exist numerical constants $\bar{c}_0, \underline{c}_1, \bar{c}_1, \bar{c}_2$ such that under parameter setting (12), $\alpha$-fraction of relevant unlabeled data and $d/n_0 > \bar{c}_0/\alpha^4$,*

$$\left(\sqrt{\frac{n_0}{d}} + \frac{72 n_0}{\alpha^2 \tilde{n}}\left(1 + \frac{\bar{c}_1}{\alpha}\left(\frac{n_0}{d}\right)^{-1/4}\right)\right)^{-1/2} \leq \frac{\mu^\top \hat{\theta}_{\text{final}}}{\sigma \left\|\hat{\theta}_{\text{final}}\right\|} \leq \left(\sqrt{\frac{n_0}{d}} + \frac{n_0}{2\alpha^2 \tilde{n}}\left(1 - \frac{\underline{c}_1}{\alpha}\left(\frac{n_0}{d}\right)^{-1/4}\right)\right)^{-1/2},$$

*with probability $\geq 1 - e^{-\bar{c}_2 \min\left\{\alpha\tilde{n}, n_0(d/n_0)^{1/4}, \sqrt{d/n_0}\right\}}$.*

The proof of Lemma 3 is technical and very similar to the proof of Lemma 2, so we defer it Section A.5.1. We remark that in the regime $\tilde{n} \geq \alpha^{-2}$, a more careful concentration argument would allow us to remove $\alpha$ from the condition $d/n_0 > \bar{c}_0/\alpha^4$ and the high order terms of the form $\frac{c_1}{\alpha}\left(\frac{n_0}{d}\right)^{-1/4}$ in Lemma 3.

Next, argue that—at least for certain values of $\mu$—the self-training estimator $\hat{\theta}_{\text{final}}$ is dense in the sense that $\|\hat{\theta}_{\text{final}}\|_1/\|\hat{\theta}_{\text{final}}\|$ is within a constant of $\sqrt{d}$.

**Lemma 4.** *Let $\mu$ be the all-ones vector. There exist constants $k_1, k_2$ such that under parameter setting (12), $\alpha$-fraction of relevant unlabeled data and $d \geq \tilde{n} \geq 30$,*

$$\frac{\|\hat{\theta}_{\text{final}}\|_1}{\|\hat{\theta}_{\text{final}}\|} \geq k_1 \sqrt{d} \text{ with probability } \geq 1 - e^{-k_0 \min\{\tilde{n}, d\}}.$$

We prove Lemma 4 in Section A.5.2. Armed with the necessary bounds, we prove Theorem 3.

*Proof of Theorem 3.* The case $\tilde{n} \geq \frac{288}{\alpha^2}\epsilon^2\sqrt{n_0 d}$ follows from Lemma 3 using an argument identical to the one used in the proof of Theorem 1. To show the case $\tilde{n} \leq \frac{c}{\alpha^2}\epsilon^2\sqrt{n_0 d}$, we take $r$ such that $\frac{c_1}{\alpha}\left(\frac{n_0}{d}\right)^{-1/4} < \frac{1}{2}$ and apply the upper bound in Lemma 3 to obtain

$$\frac{\mu^\top \hat{\theta}_{\text{final}}}{\sigma \left\|\hat{\theta}_{\text{final}}\right\|} \leq 2\alpha\sqrt{\tilde{n}/n_0} \leq \sqrt{c}\epsilon\left(\frac{d}{n_0}\right)^{1/4}$$

with probability $1 - e^{-\bar{c}_2 \min\left\{\alpha\tilde{n}, n_0(d/n_0)^{1/4}, \sqrt{d/n_0}\right\}}$. Next by Lemma 4, we have

$$\frac{\|\hat{\theta}_{\text{final}}\|_1}{\|\hat{\theta}_{\text{final}}\|} \geq \frac{\sqrt{d}}{k_1}$$

with probability at least $1 - e^{-k_0 \min\{\tilde{n}, d\}}$. Therefore, taking $c \leq \frac{1}{k_1^2}$, we have and using $\sigma = (n_0 d)^{1/4}$ and the expression (11) for the robust error probability, we have

$$\mathbb{E}_{\hat{\theta}_{\text{final}}} \text{err}_{\text{robust}}^{\infty,\epsilon}\left(f_{\hat{\theta}_{\text{final}}}\right) \geq \frac{1}{2}\mathbb{P}\left(\frac{\mu^\top \hat{\theta}_{\text{final}}}{\sigma \left\|\hat{\theta}_{\text{final}}\right\|} - \frac{\|\hat{\theta}_{\text{final}}\|_1}{\sigma\|\hat{\theta}_{\text{final}}\|} \leq 0\right)$$

$$\geq \frac{1}{2}\left(1 - e^{-\bar{c}_2 \min\left\{\alpha\tilde{n}, n_0(d/n_0)^{1/4}, \sqrt{d/n_0}\right\}} - e^{-k_0 \min\{\tilde{n}, d\}}\right).$$

Finally, we may assume without loss of generality $\alpha\tilde{n} \geq \frac{\epsilon^2\sqrt{n_0 d}}{8\log d} - n_0$ because otherwise the result holds by Theorem 1. Using $\sqrt{d/n_0} \geq \epsilon^{-2}r\log d$ and taking $r$ sufficiently large, we have that $\alpha\tilde{n} \geq \frac{\epsilon^2\sqrt{n_0 d}}{16\log d}$. Therefore, $e^{-\bar{c}_2 \min\left\{\alpha\tilde{n}, n_0(d/n_0)^{1/4}, \sqrt{d/n_0}\right\}} + e^{-k_0 \min\{\tilde{n}, d\}} \leq \frac{1}{d}$ for sufficiently large $r$. $\qquad\square$

### A.5.1 Proof of Lemma 3

The proof is largely the same as the proof of Lemma 2. We redefine $b_i$ to be the indicator of $\tilde{y}_i$ being incorrect when $i$ is relevant, and $1/2$ when it is irrelevant. Then, $\tilde{x}_i \sim \mathcal{N}\big((1-2b_i)\tilde{y}_i\mu, \sigma^2 I\big)$, and with

$$\gamma := \frac{1}{\alpha\tilde{n}}\sum_{i=1}^{\tilde{n}}(1-2b_i) \in [-1,1]$$

we may write the final classifier as

$$\hat{\theta}_{\text{final}} = \frac{1}{\tilde{n}}\sum_{i=1}^{\tilde{n}}\tilde{y}_i\tilde{x}_i = \alpha\gamma\mu + \frac{1}{\tilde{n}}\sum_{i=1}^{\tilde{n}}\tilde{y}_i\varepsilon_i$$

where $\varepsilon_i \sim \mathcal{N}(0,\sigma^2 I)$ independent of each other. Defining

$$\tilde{\delta} := \hat{\theta}_{\text{final}} - \alpha\gamma\mu$$

we have the decomposition and bound

$$\frac{\left\|\hat{\theta}_{\text{final}}\right\|^2}{\left(\mu^\top\hat{\theta}_{\text{final}}\right)^2} = \frac{1}{\|\mu\|^2} + \frac{\|\tilde{\delta}\|^2 - \frac{1}{\|\mu\|^2}\left(\mu^\top\tilde{\delta}\right)^2}{\left(\alpha\gamma\|\mu\|^2 + \mu^\top\tilde{\delta}\right)^2} \leq \frac{1}{\|\mu\|^2} + \frac{\|\tilde{\delta}\|^2}{\|\mu\|^4\left(\alpha\gamma + \frac{1}{\|\mu\|^2}\mu^\top\tilde{\delta}\right)^2}. \tag{16}$$

As argued in the proof of Lemma (2),

$$\mathbb{P}\left(\|\tilde{\delta}\|^2 \geq 2\frac{\sigma^2}{\tilde{n}}(d-1+\tilde{n})\right) \leq e^{-\tilde{n}/8} + e^{-(d-1)/8}.$$

and

$$\mathbb{P}\left(\left|\mu^\top\tilde{\delta}\right| \geq \sqrt{2}\sigma\|\mu\|\right) \leq e^{-\tilde{n}/8}.$$

Moreover, $\gamma$ is exactly the average of $1-2b_i$ over the relevant data, and therefore, as argued in Lemma (2),

$$\mathbb{P}\left(\gamma < \frac{1}{6}\right) \leq e^{-\alpha\tilde{n}/72} + e^{-c\cdot\min\left\{\sqrt{d/n_0}, n_0(d/n_0)^{1/4}\right\}}.$$

Under the event $\mathcal{E} = \left\{\|\tilde{\delta}\|^2 \leq 2\frac{\sigma^2}{\tilde{n}}(d+\tilde{n}), \left|\mu^\top\tilde{\delta}\right| \leq \sqrt{2}\sigma\|\mu\| \text{ and } \gamma \geq \frac{1}{6}\right\}$,

$$\frac{\left\|\hat{\theta}_{\text{final}}\right\|^2}{\left(\mu^\top\hat{\theta}_{\text{final}}\right)^2} \leq \frac{1}{\|\mu\|^2} + \frac{2\sigma^2(d+\tilde{n})}{\alpha^2\tilde{n}\|\mu\|^4\left(\frac{1}{6} - \frac{\sqrt{2}\sigma}{\alpha\|\mu\|}\right)^2}.$$

Substituting $\|\mu\|^2 = d$ and $\sigma^2 = \sqrt{n_0 d}$ and multiplying by $\sigma^2$, we have

$$\frac{\sigma^2\left\|\hat{\theta}_{\text{final}}\right\|^2}{\left(\mu^\top\hat{\theta}_{\text{final}}\right)^2} \leq \sqrt{\frac{n_0}{d}} + \frac{2(n_0 d)(d+\tilde{n})}{\alpha^2\tilde{n}d^2\left(\frac{1}{6} - \frac{\sqrt{2}}{\alpha}\left(\frac{n_0}{d}\right)^{1/4}\right)^2} \leq \sqrt{\frac{n_0}{d}} + \frac{72n_0}{\alpha^2\tilde{n}}\left(1 + \frac{\bar{c}_1}{\alpha}\left(\frac{n_0}{d}\right)^{1/4}\right)$$

for appropriate $\bar{c}_1$ and $\alpha^4(d/n_0)$ sufficiently large, which also implies $\mu^\top\hat{\theta}_{\text{final}} \geq 0$. To obtain the other direction of the bound, we note that in the coordinate system where the first coordinate is in the direction of $\hat{\theta}_{\text{intermediate}}$,

$$\|\tilde{\delta}\|^2 \geq \sum_{j=2}^{d}\left(\frac{1}{\tilde{n}}\sum_{i=1}^{\tilde{n}}\tilde{y}_i\varepsilon_i^{(j)}\right)^2 \sim \frac{\sigma^2}{\tilde{n}}\chi_{d-1}^2$$

and therefore

$$\mathbb{P}\left(\|\tilde{\delta}\|^2 \leq \frac{1}{2}\frac{\sigma^2}{\tilde{n}}(d-1)\right) \leq e^{-\tilde{n}/32}.$$

Therefore, under $\mathcal{E}' = \left\{ \|\tilde{\delta}\|^2 \geq \frac{1}{2}\frac{\sigma^2}{\tilde{n}}(d-1), \left|\mu^\top\tilde{\delta}\right| \leq \sqrt{2}\sigma\|\mu\| \text{ and } \gamma \geq \frac{1}{6} \right\}$, substituting into (16) we have

$$\frac{\left\|\hat{\theta}_{\text{final}}\right\|^2}{\left(\mu^\top\hat{\theta}_{\text{final}}\right)^2} \geq \frac{1}{\|\mu\|^2} + \frac{\sigma^2(d-1)}{2\alpha^2\tilde{n}\|\mu\|^4\left(1+\frac{\sqrt{2}\sigma}{\alpha\|\mu\|}\right)^2} - \frac{2\sigma^2}{\alpha^2\tilde{n}\|\mu\|^4\left(\frac{1}{6}-\frac{\sqrt{2}\sigma}{\alpha\|\mu\|}\right)^2}.$$

Substituting $\|\mu\|^2 = d$ and $\sigma^2 = \sqrt{n_0 d}$ and multiplying by $\sigma^2$, we have

$$\frac{\sigma^2\left\|\hat{\theta}_{\text{final}}\right\|^2}{\left(\mu^\top\hat{\theta}_{\text{final}}\right)^2} \geq \sqrt{\frac{n_0}{d}} + \frac{(n_0 d)(d-1)}{2\alpha^2\tilde{n}d^2\left(1+\frac{\sqrt{2}}{\alpha}\left(\frac{n_0}{d}\right)^{1/4}\right)^2} - \frac{2n_0 d}{\alpha^2\tilde{n}d^2\left(\frac{1}{6}-\frac{\sqrt{2}}{\alpha}\left(\frac{n_0}{d}\right)^{1/4}\right)^2}$$

$$\geq \sqrt{\frac{n_0}{d}} + \frac{n_0}{2\alpha^2 n}\left(1 - \frac{c_1}{\alpha}\left(\frac{n_0}{d}\right)^{1/4}\right)$$

for appropriate $c_1$ and $\alpha^4(d/n_0)$ sufficiently large. Both inequalities hold under $\mathcal{E}\cup\mathcal{E}'$, which by the preceding discussion fails with probability at most

$$e^{-\tilde{n}/32} + e^{-\alpha\tilde{n}/72} + e^{-c\cdot\min\left\{\sqrt{d/n_0},n_0(d/n_0)^{1/4}\right\}} + 2e^{-\tilde{n}/8} + e^{-(d-1)/8} \leq e^{-\bar{c}_2\min\left\{\alpha\tilde{n},n_0(d/n_0)^{1/4},\sqrt{d/n_0}\right\}}$$

for appropriate $\bar{c}_2$.

### A.5.2 Proof of Lemma 4

As in the proof of Lemma 3 we define $b_i$ to be the indicator of $\tilde{y}_i$ being incorrect when $i$ is relevant, and $1/2$ otherwise. So that we may write

$$\hat{\theta}_{\text{final}} = \alpha\gamma\mu + \frac{1}{\tilde{n}}\sum_{i=1}^{\tilde{n}}\tilde{y}_i\varepsilon_i, \text{ where } \gamma := \frac{1}{\alpha\tilde{n}}\sum_{i=1}^{\tilde{n}}(1-2b_i) \text{ and } \varepsilon_i \sim \mathcal{N}\left(0,\sigma^2 I\right).$$

We further decompose $\varepsilon_i$ to components orthogonal and parallel to $\hat{\theta}_{\text{intermediate}}$, $\varepsilon_i = \varepsilon_i^\perp + \varepsilon_i^\parallel$, so that $\varepsilon_i^\perp \sim \mathcal{N}\left(0,\sigma^2\left(1-\pi\pi^T\right)\right)$ and $\varepsilon_i^\parallel \sim \mathcal{N}\left(0,\sigma^2\pi\pi^T\right)$, where $\pi$ is a unit vector parallel to $\hat{\theta}_{\text{intermediate}}$. We note that $\varepsilon_i^\perp$ is independent of $\tilde{y}_i$ and of $\varepsilon_i^\parallel$. Let $[v]_j$ denote the $j$th coordinate of vector $v$ in the standard basis, such that

$$\|\hat{\theta}_{\text{final}}\|_1 = \sum_{j=1}^{d}|[\hat{\theta}_{\text{final}}]_j| = \sum_{j=1}^{d}\left|\alpha\gamma + \frac{1}{\tilde{n}}\sum_{i=1}^{\tilde{n}}\tilde{y}_i\left[\varepsilon_i^\perp\right]_j + \frac{1}{\tilde{n}}\sum_{i=1}^{\tilde{n}}\tilde{y}_i\left[\varepsilon_i^\parallel\right]_j\right|.$$

We define

$$J = \left\{j \mid [\pi]_j^2 \leq \frac{2}{d}\right\}, \text{ so that } |J| \geq \frac{d}{2}$$

since $\pi$ is a unit vector. For every $j$, $\frac{1}{\tilde{n}}\sum_{i=1}^{\tilde{n}}\tilde{y}_i\left[\varepsilon_i^\perp\right]_j \sim \mathcal{N}\left(0,\frac{\sigma^2}{\tilde{n}}\left(1-[\pi]_j^2\right)\right)$ and therefore for every $j \in J$

$$\mathbb{P}\left(\frac{1}{\tilde{n}}\sum_{i=1}^{\tilde{n}}\tilde{y}_i\left[\varepsilon_i^\perp\right]_j > \frac{\sigma}{\sqrt{\tilde{n}}}\left(1-\frac{2}{d}\right)\right) \geq Q(1) \geq \frac{1}{8}.$$

Moreover, by Cauchy–Schwarz

$$\left(\frac{1}{\tilde{n}}\sum_{i=1}^{\tilde{n}}\tilde{y}_i\left[\varepsilon_i^\parallel\right]_j\right)^2 \leq \frac{1}{\tilde{n}}\sum_{i=1}^{\tilde{n}}\left[\varepsilon_i^\parallel\right]_j^2 \sim \frac{\sigma^2[\pi]_j^2}{\tilde{n}}\chi_{\tilde{n}}$$

and therefore for every $j \in J$

$$\mathbb{P}\left(\left|\frac{1}{\tilde{n}}\sum_{i=1}^{\tilde{n}}\tilde{y}_i\left[\varepsilon_i^\parallel\right]_j\right| > \frac{2\sigma}{\sqrt{\tilde{n}d}}\right) \leq e^{-\tilde{n}/8}.$$

Therefore, with probability at at least $Q(1) - e^{-\tilde{n}/8} \geq \frac{1}{10}$ for $\tilde{n} \geq 30$,

$$\left| \alpha\gamma + \frac{1}{\tilde{n}} \sum_{i=1}^{\tilde{n}} \tilde{y}_i \left[ \varepsilon_i^\perp \right]_j + \frac{1}{\tilde{n}} \sum_{i=1}^{\tilde{n}} \tilde{y}_i \left[ \varepsilon_i^\| \right]_j \right| \geq \alpha\gamma + \frac{\sigma}{\sqrt{\tilde{n}}} \left( \sqrt{1 - \frac{2}{d}} - \frac{2}{\sqrt{d}} \right) \geq \alpha\gamma + \frac{\sigma}{2\sqrt{\tilde{n}}} \qquad (17)$$

for $d \geq 20$. We implicitly assumed here $\gamma > 0$, which we have previously argued to hold with high probability. However, an analogous bound holds if $\gamma \leq 0$ and so we don't need to take the this into account here.

Let $J'$ be the random set of coordinates for which the inequality (17) holds; we have $\|\hat{\theta}_{\text{final}}\|_1 \geq d\left(\alpha\gamma + \frac{\sigma^2}{2\tilde{n}}\right)|J'| \geq d\left(\alpha\gamma + \frac{\sigma^2}{2\tilde{n}}\right)|J' \cap J|$. Moreover, by the above discussion $|J' \cap J|$ is binomial with at least $d/2$ trials and success probability at least $1/10$. Therefore

$$\mathbb{P}\left( |J' \cap J| \leq \frac{d}{20} - \frac{d}{40} \right) \leq e^{-2(d/2)/20^2} = e^{-d/400}.$$

And consequently we have

$$\|\hat{\theta}_{\text{final}}\|_1 \geq \frac{d}{40}\left( \alpha\gamma + \frac{\sigma^2}{2\tilde{n}} \right) \text{ with probability} \geq 1 - e^{-d/400}.$$

Next we need to argue about the Euclidean norm

$$\|\hat{\theta}_{\text{final}}\| = \|\alpha\gamma\mu + \tilde{\delta}\| \leq \|\alpha\gamma\mu\| + \|\tilde{\delta}\| = \alpha\gamma\sqrt{d} + \|\tilde{\delta}\|$$

with $\tilde{\delta} = \frac{1}{\tilde{n}}\sum_{i=1}^{\tilde{n}} \tilde{y}_i \varepsilon_i$. As argued in Eq. (15) in the proof of Lemma 2,

$$\mathbb{P}\left( \|\tilde{\delta}\|^2 \geq 2\frac{\sigma^2}{\tilde{n}}(d - 1 + \tilde{n}) \right) \leq e^{-\tilde{n}/8} + e^{-(d-1)/8}.$$

Under $\|\hat{\theta}_{\text{final}}\|_1 \geq \frac{d}{40}\left( \alpha\gamma + \frac{\sigma^2}{2\tilde{n}} \right)$ and $\|\tilde{\delta}\|^2 \leq 2\frac{\sigma^2}{\tilde{n}}(d - 1 + \tilde{n}) \leq 2\frac{\sigma^2}{\tilde{n}}\left( \sqrt{d} + \sqrt{\tilde{n}} \right)^2$ we have

$$\frac{\|\hat{\theta}_{\text{final}}\|_1}{\|\hat{\theta}_{\text{final}}\|} \geq \frac{d}{40\sqrt{d}} \frac{\alpha\gamma + \frac{1}{2}\sigma\tilde{n}^{-1/2}}{\alpha\gamma + \sqrt{2}\sigma\tilde{n}^{-1/2} + \sqrt{2}\sigma d^{-1/2}} \geq \frac{\sqrt{d}}{k_1}$$

for $d \geq \tilde{n}$ and $k_1 = 160\sqrt{2}$. By the preceding discussion, this happens with probability at least $1 - e^{-\tilde{n}/8} - e^{-(d-1)/8} - e^{-d/400} \geq 1 - e^{k_1 \min\{\tilde{n}, d\}}$.

## B  CIFAR-10 experimental setup

### B.1  Training hyperparameters

Here we describe the training hyperparameters used in our main CIFAR-10 experiments. Our additional experiments in Appendix C use the same hyperparameters unless otherwise mentioned.

**Architecture.**  We use a Wide ResNet 28-10 architecture, as in [29] and similarly to [56], who use a 34-10 variant.

**Robust self-training.**  We set the regularization weight $\beta = 6$ as in [56]. We implicitly set the unlabeled data weight to $w = 50\text{K}/500\text{K} = 0.1$ by composing every batch from equal parts labeled and unlabeled data.

**Adversarial self-training.**  We compute $x_{\text{PG}}$ exactly as in [56], with step size $0.007$, 10 iterations and $\epsilon = 8/255$.

**Stability training.**  We set the additive noise variance to $\sigma = 0.25$. We perform the certification using the randomized smoothing protocol described in [9], with parameters $N_0 = 100$, $N = 10^4$, $\alpha = 10^{-3}$ and noise variance $\sigma = 0.25$.

**Input normalization.**  We scale each pixel in the input image by $1/255$, to be in $[0,1]$.

**Data augmentation.**    We perform the standard CIFAR-10 data augmentation: a random 4-pixel crop followed by a random horizontal flip.

**Optimizer configuration.**    We use the hyperparameters [10] prescribe for Wide ResNet 28-10 and CIFAR-10, except for batch size and number of epochs: initial learning rate $0.1$, cosine learning rate annealing [28] (without restarts), weight decay $5 \cdot 10^{-4}$ and SGD with Nesterov momentum $0.9$. To speed up robust training, we doubled the batch size from 128 and 256. (This way, every batch has 128 original CIFAR-10 images and 128 pseudo-labeled imaged).

**Number of gradient steps.**    Since we increase the dataset size by a factor of 10, we expect to require more steps for training to converge. However, due to the high cost of adversarial training (caused by the inner optimization loop), we restricted the training of $\texttt{RST}_{\texttt{adv}}(\texttt{50K+500K})$ to 39K gradient steps. This corresponds to 100 CIFAR-10 epochs at batch size 128, which is standard. Stability training is much cheaper (involving only random sampling at each step), and we train $\texttt{RST}_{\texttt{stab}}(\texttt{50K+500K})$ for 156K gradient steps. Training for longer will likely enhance performance, but probably not dramatically.

**Pseudo-label generation.**    We used the same model to generate the pseudo-labels in all of our CIFAR-10 robust-training experiments. They were generated by a Wide ResNet 28-10 which we trained on the CIFAR-10 training set only. The training parameters were exactly like those of the baseline in [10] (with standard augmentation). That is to say, training parameters were as above, except we used batch size 128 and ran 200 epochs. The resulting model had 96.0% accuracy on the CIFAR-10 test set.

**Baselines.**    For fully supervised adversarial training the above hyperparameter configuration fails due to overfitting, as we report in detail in Appendix B.4. Stability training did not exhibit overfitting, but longer training did not improve results: we trained $\texttt{Baseline}_{\texttt{st}}(\texttt{50K})$ for 19.5K gradient steps, i.e. 100 epochs at batch size 256. We also tried training for 200 and 400 epochs, but saw no improvement when comparing certification results over 10% of the data.

## B.2   Implementation and running times

We implement our experiments in PyTorch [35], using open source code from [56, 9]. We ran all our experiments on Titan Xp GPU's. Training $\texttt{RST}_{\texttt{adv}}(\texttt{50K+500K})$ took 28 hours on 4 GPU's. Running $\texttt{PG}_{\texttt{Ours}}$ on a Wide ResNet 28-10 took 30 minutes on 4 GPU's. Training $\texttt{RST}_{\texttt{stab}}(\texttt{50K+500K})$ took 40 hours on 2 GPU's. Running randomized smoothing certification on a Wide ResNet 28-10 took 19 hours on 2 GPU's.

## B.3   Tuning attacks

In this section, we provide details on how we tuned the parameters for the projected gradient attack to evaluate adversarially trained models. Let $\texttt{PG}(\eta,\tau,\rho)$ to denote an attack with step-size $\eta$ performing $\tau$ steps, with $\rho$ restarts. We tune the attacks to maximally reduce the accuracy of our model $\texttt{RST}_{\texttt{adv}}(\texttt{50K+500K})$.

For every restart, we start with a different random initialization within the $\ell_\infty$ ball of radius $\epsilon$. At every step of every restart, we check if an adversarial example (i.e. an input that causes misclassification in the model) was found. We report the final accuracy as the percentage of examples where no successful adversarial example was found across all steps and restarts.

We first focus on $\epsilon = 8/255$ which is the main size of perturbation of interest in this paper. We report all the numbers with 1 significant figure and observe around $0.05\%$ variation across multiple runs of the same attack on the same model.

**Step-size.**    We use $\rho = 5$ restarts and experiment tune the number of steps and the steps size. We tried $20, 40, 60$ steps and step sizes $0.005$, $0.01$ and $0.02$. Table 2 summarizes the results. We see that the mid step-size $\eta = 0.01$ provided the best accuracy, across the range of steps. We also chose $\tau = 40$ for computational benefit, since larger $\tau$ did not seem to provide much gain. We thus obtained the $\texttt{PG}_{\texttt{Ours}}$ configuration with $\eta = 0.01, \tau = 40, \rho = 5$, that we used to test multiple reruns of our model and other models from the literature.

To compare to previous attacks typically used in the literature: $\texttt{PG}_{\texttt{Madry}}$ [29] corresponds to $\eta = 0.007$, $\rho = 1$ and $\tau = 20$, and $\texttt{PG}_{\texttt{TRADES}}$ [56] corresponds to $\eta = 0.003$, $\rho = 1$ and $\tau = 20$, *without random*

| Number of steps $\tau$ | $\eta=0.005$ | $\eta=0.01$ | $\eta=0.02$ |
|:---:|:---:|:---:|:---:|
| $\tau=20$ | 63.4 | **62.9** | 62.9 |
| $\tau=40$ | 62.8 | **62.5** | 63 |
| $\tau=60$ | 62.6 | **62.5** | 62.8 |

Table 2: Tuning the step-size $\eta$ for the PG attack. Over the range of step numbers considered, $\eta=0.01$ was the most effective against our model.

| Number of restarts | Robust accuracy |
|:---:|:---:|
| $\rho=2$ | 62.7 |
| $\rho=4$ | 62.5 |
| $\rho=6$ | 62.5 |
| $\rho=8$ | 62.4 |
| $\rho=10$ | 62.4 |
| $\rho=12$ | 62.4 |
| $\rho=14$ | 62.3 |
| $\rho=16$ | 62.3 |
| $\rho=18$ | 62.3 |
| $\rho=20$ | 62.3 |

Table 3: Effect of number of restarts on robust accuracy with fixed step-size $\eta=0.01$ and number of steps $\tau=40$.

*initializations* (attack is always initialized at the test input $x$). We use both more steps and more restarts than used typically, and also tune the step-size to observe that $\eta=0.01$ was worse for our model than the smaller step-size of $0.007$.

**Number of restarts.** PG attack which performs projected gradient method on a non-convex objective is typically sensitive to the exact initialization point. Therefore, multiple restarts are generally effective in bringing down robust accuracy. We now experiment with the effect of number of restarts. We perform upto 20 restarts and see a very gradual decrease in robust accuracy, with around $0.2\%$ drop from just using 5 restarts. See Table 3 for the robust accuracies at different number of restarts. We remark that using a much larger number of steps or restarts could cause additional degradation (as in [18]), but we stick to the order of steps and restarts that are typically reported in the literature.

**Fine-tuning for different $\epsilon$.** Figure 16 reports the accuracies for different $\epsilon$. A smaller $\epsilon$ would typically require a smaller step-size. We fine-tune the step-size for each $\epsilon$ separately, by fixing the number of restarts to 5 and number of steps to $40$.

- For $\epsilon=0.008$, we span $\eta\in\{0.001,0.002,0.005\}$.

- For $\epsilon=0.016$, we span $\eta\in\{0.002,0.005,0.01\}$.

- For $\epsilon=0.024$, we span $\eta\in\{0.005,0.01,0.02\}$.

- For $\epsilon=0.039$, we span $\eta\in\{0.005,0.01,0.02\}$.

## B.4   Comparison with hyperparameters in [56]

As a baseline for adversarial robust self-training, we attempted to reproduce the results of [56], whose publicly-released model has $56.6\%$ robust accuracy against $\text{PG}_{\text{TRADES}}$, $55.3\%$ robust accuracy against $\text{PG}_{\text{Ours}}$, and $84.9\%$ standard accuracy. However, performing adversarial training with the hyper-parameters described in Appendix B.1 produces a poor result: the resulting model has only $50.8\%$ robust accuracy against $\text{PG}_{\text{Ours}}$, and slightly better $85.8\%$ standard accuracy. We then changed all the hyper-parameters to be the same as in [56], with the exception of the model architecture, which we kept at Wide ResNet 28-10. The resulting model performed somewhat better, but still not on par with the numbers reported in [56].

Figure 4: Comparison of training traces for different hyperpameters of adversarial training. Dashed lines show standard accuracy on the entire CIFAR-10 test set, and whole lines show robust accuracy against $PG_{TRADES}$ evaluated on the first 500 images in the CIFAR-10 test set.

Examining the training traces (Figure 4) reveals that without unlabeled data, both hyper-parameter configurations suffer from overfitting. More precisely, the robust accuracy degrades towards the end of the training, while standard accuracy gradually improves. In contrast, we see no such overfitting with $RST_{adv}$(50K+500K), directly showing how unlabeled data aids generalization.

Finally, we perform "early-stopping" with the model trained according to [56], selecting the model with highest validation robust accuracy. This model has 55.5% robust accuracy against $PG_{TRADES}$, 54.1% robust accuracy against $PG_{Ours}$, and 84.5% standard accuracy. This is reasonably close to the result of [56], considering we used a slightly lower-capacity model.

## B.5   Comparison between stability and noise training

As we report in Section 5.1.2, our fully supervised baseline for stability training, i.e. $Baseline_{stab}$(50K), significantly outperforms the model trained in [9] and available online. There are three a-priori reasons for that: (i) our Wide ResNet 28-10 has higher capacity than the ResNet-110 used in [9], (ii) we employ a different training configuration (e.g. a cosine instead of step learning rate schedule) and (iii) we use stability training while Cohen et al. [9] use a different training objective. Namely, Cohen et al. [9] add $\mathcal{N}(0,\sigma^2 I)$ to the input during training, but treat the noise as data augmentation, minimizing the loss

$$\mathbb{E}_{x'\sim\mathcal{N}(x,\sigma^2 I)}L_{standard}(\theta,x',y).$$

We refer to the training method of [9] as *noise training*.

To test which of these three differences causes the gap in performance we train the following additional models. First, we perform noise training with ResNet-110, but otherwise use the same configuration used to train $Baseline_{stab}$(50K). Second, we keep the ResNet-110 architecture and our training configuration, and use stability training instead. Finally, we perform noise training on Wide ResNet 28-10 with all other parameters the same as $Baseline_{stab}$(50K). As our goal in this section is to compare supervised training techniques for randomized smoothing, these experiments only use the CIFAR-10 training set (and no unlabeled data).

We plot the performance of all of these models, as well as the model of [9] and $Baseline_{stab}$(50K), in Figure 5. Starting with the model of [9] and using our training configuration increases accuracy by 2–3% across most perturbation radii. Switching to stability training reduces clean accuracy by roughly 3%, but dramatically increases robustness at large radii, with a 13% improvement at radius 0.5. Using the larger Wide ResNet 28-10 model further improves performance by roughly 2%. We also see that noise training on Wide ResNet 28-10 performs better at radii below 0.25, and worse on larger radii. With stability training it is possible to further explore the tradeoff between accuracy at low and high radii by tuning the parameter $\beta$ in (6), but we did not pursue this (all of our experiments are with $\beta=6$).

Figure 5: Certified accuracy as a function of $\ell_2$ perturbation radius, comparing two architectures, two hyperparameter sets and two training objectives. **(a)** Both stability training and our hyperparameter choice improve performance. **(b)** Increasing model capacity improves performance further, and stability training remains beneficial.

## B.6 Sourcing unlabeled data for CIFAR-10

Here we provide a detailed description of the sourcing process we describe in Section 5.1.1. To obtain unlabeled data distributed similarly to the CIFAR-10 images, we use the 80 Million Tiny Images (80M-TI) dataset [46]. This dataset contains 79,302,017 color images that were obtained via querying various keywords in a number of image search engines, and resizing the results to a 32x32 resolution. CIFAR-10 is a manually-labeled subset of 80M-TI. However, most of the 80M-TI do not fall into any of the CIFAR-10 categories (see Figure 6a) and the query keywords constitute very weak labels (Figure 6b). To select relevant images, we train a classifier to classify TI data as relevant or not, in the following steps.

**Training data for selection model.** We create an 11-class training set consisting of the CIFAR-10 training set and 1M images sampled at random from the 78,712,306 images in 80M-TI with keywords that did not appear in CIFAR-10. We similarly sample an additional 10K images for validation.

**Training the selection model.** We train an 11-way classifier on this dataset, with the same Wide ResNet 28-10 architecture employed in the rest of our experiments. We use the hyperparmeters described in Appendix B.1, except we run for 117K gadient steps. We use batch size 256 and comprise each batch from 128 CIFAR-10 images and 128 80M-TI images, and we also weight the loss of the "80M-TI" class by 0.1 to balance its higher number of examples. During training we evaluate the model on its accuracy of discriminating between CIFAR-10 and 80M-TI on a combination of the CIFAR-10 test and the 10K 80M-TI validation images, and show the training trace in Figure 8. Towards the end of the training, the CIFAR-10 vs. 80M-TI accuracy started to degrade, and we therefore chose and earlier checkpoint (marked in the figure) to use as the data selection model. This model achieves 93.8% CIFAR-10 vs. 80M-TI test accuracy.

**Removing CIFAR-10 test set.** To ensure that there is no leakage of the CIFAR-10 test set to the unlabeled data we source, we remove from 80M-TI all near-duplicates of the CIFAR-10 test set. Following [39], we define a near-duplicate as an image with $\ell_2$ distance below $2000/255$ to any of the CIFAR-10 test images. We visually confirm that images with distance greater than $2000/255$ are substantially different. Our near-duplicate removal procedure leaves 65,807,640 valid candidates.

**Selecting the unlabeled data.** We apply our classifier on 80M-TI, with all images close to the CIFAR-10 test set excluded as described above. For each CIFAR-10 class, we select the 50,000 images which our classifier predicts with the highest confidence as belonging to that class. This is our unlabeled dataset, depicted in Figure 7, which is 10x the original CIFAR-10 training set and approximately class balanced.

Examining Figure 7, it is clear that our unlabeled dataset is not entirely relevant to the CIFAR-10 classification task. In particular, many of the "frog" and "deer" images are not actually frogs and deer. It is likely possible to obtain higher quality data by tuning the selection model (and particularly its training) more carefully. We chose not to do so for two reasons. First, allowing some amount of irrelevant unlabeled data more realistically simulates robust self-training in other contexts. Second,

Figure 6: Random images from the 80 Million Tiny Images data. **(a)** Images drawn from the entire dataset. **(b)** Images drawn for the subset with keywords that appeared in CIFAR-10; matching keywords correlate only weakly with membership in one of the CIFAR-10 classes.

Figure 7: A random sample of our approximately class-balanced 500K auxiliary images, rows correspond to class predictions made by our data selection model. Note the multiple errors on "frog" and "deer."

for a totally fair comparison against [56], we chose not to use state-of-the-art architectures or training techniques for the data selection model, and instead make it as close as possible to the final robust model.

## C  Additional CIFAR-10 experiments

### C.1  Alternative semisupervised training method

In this section, we consider the straightforward adaptation of *virtual adversarial training* (VAT) to the (real) adversarial robustness setting of interest in this paper.

**Training objective.**   Recall that we consider robust losses on the following form.

$$L_{\text{robust}}(\theta,x,y) = L_{\text{standard}}(\theta,x,y) + \beta L_{\text{reg}}(\theta,x),$$
$$\text{where }\ L_{\text{reg}}(\theta,x) := \max_{x' \in \mathcal{B}_\epsilon^p(x)} D_{\text{KL}}(p_\theta(\cdot\,|\,x)\,\|\,p_\theta(\cdot\,|\,x')).$$

The $L_{\text{reg}}$ term does not require labels, and hence could be evaluated on the unlabeled data directly—exactly as done in Virtual Adversarial Training. As is commonly done in standard semisupervised

Figure 8: Test accuracy through training for the unlabeled data selection model.

| Model architecture | Training algorithm | $\text{PG}_{\text{Ours}}$ | No attack |
|---|---|---|---|
| Wide ResNet 40-2 | $\texttt{Baseline}_{\text{adv}}(50\text{K})$ | 52.1 | 81.3 |
| Wide ResNet 40-2 | $\texttt{rVAT}_{\text{adv}}(50\text{K}+500\text{K}), \lambda_{\text{ent}}=0$ | 53.6 | 81.5 |
| Wide ResNet 40-2 | $\texttt{rVAT}_{\text{adv}}(50\text{K}+500\text{K}), \lambda_{\text{ent}}=0.1$ | 52.9 | 77.9 |
| Wide ResNet 40-2 | $\texttt{rVAT}_{\text{adv}}(50\text{K}+500\text{K}), \lambda_{\text{ent}}=0.6$ | 43.4 | 60.0 |
| Wide ResNet 40-2 | $\texttt{rVAT}_{\text{adv}}(50\text{K}+500\text{K}), \lambda_{\text{ent}}=3.0$ | 15.4 | 19.1 |
| Wide ResNet 28-10 | TRADES [56] | 55.4 | 84.9 |
| Wide ResNet 28-10 | $\texttt{rVAT}_{\text{adv}}(50\text{K}+500\text{K}), \lambda_{\text{ent}}=0$ | 56.5 | 83.2 |
| Wide ResNet 28-10 | $\texttt{RST}_{\text{adv}}(50\text{K}+500\text{K})$ | 62.5 | 89.7 |

Table 4: Accuracy of adversarially trained models against our PG attack. We see that VAT-like consistency-based regularization produces only minor gains over a baseline without unlabeled data, significantly underperforming robust self-training.

learning, we also consider an additional entropy regularization term, to discourage the model from reaching the degenerate solution of mapping unlabeled inputs to uniform distributions. The total training objective is

$$\sum_{i=1}^{n} L_{\text{standard}}(\theta, x_i, y_i) + \beta\left(\sum_{i=1}^{n} L_{\text{reg}}(\theta, x_i) + w \sum_{i=1}^{\tilde{n}} L_{\text{reg}}(\theta, \tilde{x}_i)\right) + \lambda_{\text{ent}} \sum_{i=1}^{\tilde{n}} h(p_\theta(\cdot \mid \tilde{x}_i)), \quad (18)$$

where $h(p_\theta(\cdot \mid \tilde{x}_i)) = -\sum_{y \in \mathcal{Y}} p_\theta(y \mid \tilde{x}_i) \log p_\theta(y \mid \tilde{x}_i)$ is the entropy of the probability distribution over the class labels. We denote models trained according to this objective by $\texttt{rVAT}$.

There are two differences between VAT and (18). First, to minimize the loss, VAT takes gradients of $D_{\text{KL}}(p_\theta(\cdot \mid x) \| p_\theta(\cdot \mid x'))$ w.r.t. $\theta$ only through $p_\theta(\cdot \mid x')$, treating $p_\theta(\cdot \mid x)$ as a constant. Second, VAT computes perturbations $x' \in \mathcal{B}_\epsilon^2(x)$ that are somewhere between random and adversarial, using a small number of (approximate) power iterations on $\nabla_{x'}^2 D_{\text{KL}}(p_\theta(\cdot \mid x) \| p_\theta(\cdot \mid x'))$.

We experiment with both the adversarial training based regularization (7) and stability training based regularizer (8).

**Training details.** We consider the alternative semisupervised objective (18) and compare with robust self-training. We use the same hyperparmeters as the rest of the main experiments, described in Appendix B.1, and tune the additional hyperparameter $\lambda_{\text{ent}}$. Setting $\lambda_{\text{ent}} = \beta = 6$ would correspond to the entropy weight suggested in VAT [30], since they use a logarithmic loss and not KL-divergence. We experiment with $\lambda_{\text{ent}}$ between 0 and 6. Due to computational constraints, we use the smaller model Wide ResNet 40-2 for tuning this hyperparameter in the adversarially trained models. We also perform certification via randomized smoothing on 1000 random examples from the test set, rather than the entire test set.

Figure 9: Certified $\ell_2$ accuracy as a function of the radius, for Wide ResNet 28-10 trained using different semisupervised approaches on the augmented CIFAR10 dataset.

Figure 10: Effect of entropy weight on performance of stability training with alternate semisupervised approach. Larger entropy weight leads to gains in robustness at larger radii at a cost in robustness at smaller radii. However, robust self-training outperforms the alternative semisupervised approach over all radii, for all settings of the entropy weight hyperparameter.

**Results.** Figures 9 and 10 summarize the results for stability training. We see that the alternative approach also yields gains (albeit much smaller than robust self-training) over the baseline supervised setting, due to the additional unlabeled data.

Table 4 presents the accuracies against $\text{PG}_{\text{Ours}}$ for the smaller Wide ResNet 40-2 models with different settings of $\lambda_{\text{ent}}$. We see a steady degradation in the performance with increasing $\lambda_{\text{ent}}$. For $\lambda_{\text{ent}} = 0$, we also train a large Wide ResNet 28-10 to compare to our state-of-the-art self-trained $\text{RST}_{\text{adv}}(50\text{K}+500\text{K})$ that has the same architecture.

We see that in both adversarial training (heuristic defense) and stability training (certified defense), robust self-training significantly outperforms the alternative approach suggesting that "locking in" the pseudo-labels is important. This is possibly due to the fact that we are in a regime where the pseudo-labels are quite accurate and hence provide good direct signal to the model. Another possible explanation for the comparative weak performance of rVAT is that since the robustly-trained never reaches good clean accuracy, it cannot effectively bootstrap the model's own prediction as training progresses.

**Additional experiments.** VAT combined with entropy minimization is one of the most succesful semisupervised learning algorithm for standard accuracy [33]. As we mentioned earlier, the objective in (18) differs from VAT by not treating $p_\theta(\cdot \mid \tilde{x})$ as a constant. From the open source implementation[4], we also note that batch normalization parameters are not updated on the "adversarial" perturbations during training. We tried both variants on the smaller model Wide ResNet 40-2 and observed that neither modification affected final performance in our setting. Further, we also experimented with different values of the unlabeled data weighting parameter $w$. We observed no noticeable improvement in final performance by changing this parameter.

| Model | PG$_{\text{Ours}}$ | No attack |
|---|---|---|
| TRADES [56] | 55.4 | 84.9 |
| Baseline$_{\text{adv}}$(50K) | 47.5 | 84.8 |
| + Cutout [13] | 51.2 | 85.8 |
| + AutoAugment [10] | 47.4 | 84.5 |
| RST$_{\text{adv}}$(50K+500K) | 62.5 | 89.7 |

Table 5: Accuracy of adversarially trained models against PG$_{\text{Ours}}$. We see that cutout provides marginal gains and AutoAugment leads to slightly worse performance than our baseline that only uses the standard crops and flips.

Figure 11: Comparing certified $\ell_2$ accuracy as a function of certified radius, computed via randomized smoothing for different data augmentation methods.

### C.2 Comparison with data augmentation

Advanced data augmentation techniques such as cutout [13] and AutoAugment policies [10] provide additional inductive bias about the task at hand. Can data augmentation provide gains in CIFAR-10 robust training, similar to those we observe by augmenting the CIFAR-10 dataset with extra unlabeled images?

**Implementation details.** We use open source implementations for cutout[5] and AutoAugment[6], and use the same training hyperparameters from the papers introducing these techniques. We first train Wide ResNet 28-10 via standard training and reproduce the test accuracies reported in the papers [13, 10].

**Training details.** We perform robust supervised training, where the each batch contains *augmented* (cutout/autoaugment in additional to random crops and flips) of the CIFAR10 training set.

We use the same training setup with which we performed robust self-training, as described in Appendix B.1, except we applied augmentation instead of adding unlabeled data, and that we perform stability training for only 39K steps. Note that this model and optimization configuration are identical to those used in the AutoAugment paper [10], except increasing batch size to 256.

**Results.** Table 5 presents the results on accuracy of the heuristic adversarially trained models against our PG variant PG$_{\text{Ours}}$ for $\epsilon=8/255$. We also tabulate the performance of Baseline$_{\text{adv}}$(50K), which doesn't use any unlabeled data or augmentation—as we show in Appendix B.4, this configuration produces poor results due to overfitting. We see that AutoAugment offers no improvement over Baseline$_{\text{adv}}$(50K), while cutout ofer a 4% improvement that is still far from the performance [56] attains with just early stopping. In Figure 12 we plot training training traces, and show that AutoAugment fails to prevent overffiting, while cutout provides some improvement, but is till far away from the effect of robust self-training.

We also perform certification using randomized smoothing on the stability trained models. We use the same certification parameters as our main experiments ($N_0=100, N=10^4, \alpha=10^{-3}, \sigma=0.25$) and

Figure 12: Training traces adversarial training with robust self-training and various forms of data augmentation. Dashed lines show standard accuracy on the entire CIFAR-10 test set, and whole lines show robust accuracy against $\mathrm{PG_{TRADES}}$ evaluated on the first 500 images in the CIFAR-10 test set. AutoAugment shows the same overfitting as standard augmentation, while cutout mitigates overfitting, but does not provide the gains of self-training.

compute the certificate over every 10th image in the CIFAR-10 test set (1000 images in all) for all the models. As we only train for 31K steps, we compare the results to an $\mathrm{RST_{stab}}(50K{+}500K)$ model trained for the same amount of steps (and evaluated on the same subset of images). We plot the results to obtain Figure 11. As can be seen, data augmentation provides performance essentially identical to that of a baseline without augmentation. Overall, it is interesting to note that the gains provided by data augmentation for standard training do not manifest in robust training (both adversarial and stability training).

### C.3 Effect of unlabeled data relevance

Semisupervised learning algorithms are often sensitive to the relevance of the unlabeled data. That is, if too much of the unlabeled data does not correspond to any label, the learning algorithm might fail to benefit from the unlabeled dataset, and may even produce worse results simply using labeled data only. This sensitivity was recently demonstrated in a number of semisupervised neural network training techniques [33]. In the simple Gaussian model of Section 3, our analysis in Appendix A.5 shows that any fixed fraction of relevant unlabeled data will allow self-training to attain robustness, but that the overall sample complexity grows inversely with the square of the relevant fraction.

To test the practical effect of data relevance on robust self-training, we conduct the following experiment. We draw 500K random images from the 80M-TI with CIFAR-10 test set near-duplicates removed (see Appendix B.6), which rarely portray one of the CIFAR-10 classes (see Figure 6a); this is our proxy for irrelevant data. We mix the irrelevant dataset and our main unlabeled image dataset with different proportions, creating a sequence of unlabeled datasets with a growing degree of image relevance, each with 500K images.

We then perform adversarial- and stability- robust self-training on each of those datasets as described in Section 5 and Appendix B.1, except here we use a smaller Wide ResNet 40-2 model to conserve computation resources. We evaluate each of those models as in Section 5.1.2, and compare them to a fully supervised baseline. For stability training we train the baseline as in Section 5.1.2, except with Wide ResNet 40-2. For adversarial training, in view of our findings in Appendix B.4, we train our baseline with Wide ResNet 40-2 and the training hyperparameters of [56] ($\beta$ is still 6). Here, we did not observe overfitting in the training trace, and therefore used the model from the final epoch of training. This baseline achieves 81.3% standard accuracy and 52.1% robust accuracy against $\mathrm{PG_{Ours}}$.

In Figure 13 we plot the difference in $\mathrm{PG_{Ours}}$/certified accuracy as a function of the relevant data fraction. We see the performance grows essentially monotonically with data relevance, as expected. We also observe that smoothing seems to be more sensitive to data relevance, and we see performance degradation for completely irrelevant and 20% relevant data. For adversarial training we see no performance degradation, but the gain with completely irrelevant data is negligible as can be expected.

Figure 13: Performance improvement as a function of unlabeled data relevance, for Wide ResNet 40-2 models, relative to training without unlabeled data. When present, error bars indicate the range of variation (minimum to maximum) over 5 independent runs. **(a)** Difference in standard accuracy and in accuracy under the $\ell_\infty$ attack `PG_Ours`, for adversarially-trained models. **(b)** Difference in standard accuracy and in certified $\ell_2$ accuracy for smoothed stability-trained models.

Finally, at around 80% relevant data performance seems close to that of 100% relevant data. This is perhaps not too surprising, considering that even our "100% relevant" data is likely somewhere between 90-95% relevant (see Appendix B.6). This is also roughly in line with our theoretical model, where there is a small degradation in performance for relevant fraction $\alpha = 0.8$.

### C.4   Effect of unlabeled data amount

We have conducted all of our previous experiments on an unlabeled dataset of fixed size (500K images). Here we test the effect of the size of the unlabeled dataset on robust self-training performance. The main question we would like to answer here is how much further performance gain can we expect from further increasing the data set: should we expect another 7% improvement over [56] by adding 500K more images (of similar relevance), or perhaps we would need to add 5M images, or is it the case that the benefit plateaus after a certain amount of unlabeled data?

There is a number of complicating factors in answering this question. First, 80M-TI does not provide us much more than 500K relevant unlabeled images, certainly not if we wish it to be approximately class-balanced; the more images we take from 80M-TI the lower their relevance. Second, as the amount of data changes, some training hyperparameters might have to change as well. In particular, with more unlabeled data, we expect to require more gradient steps before achieving convergence. Even for 500K images we haven't completely exhausted the possible gains from longer training. Finally, it might be the case that higher capacity models are crucial for extracting benefit from larger amounts of unlabeled data. For these reasons, and considering our computational constraints, we decided that attempting to rerun our experiment with more unlabeled data will likely yield inconclusive results, and did not attempt it.

Instead, we sub-sampled our 500K images set randomly into nested subsets of varying size, and repeated our experiment of Section 5.1.2 with these smaller datasets. With this experiment we hope to get some understanding of the trend with which performance improves. Since we expect model capacity to be very important for leveraging more data, we perform this experiment using the same high-capacity Wide ResNet 28-10 used in our main experiment. For adversarial training, we use exactly the same training configuration for all unlabeled dataset sizes. For stability training, we also use the same configuration except we attempted to tune the number of gradient steps. For each dataset size, we started by running 19.5K gradient steps, and doubled the number of gradient steps until we no longer saw improvement in performance. For 40K extra data, we saw the best results with 39K gradient steps, and for 100K and 240K extra data we saw the best result with 78K gradient steps. Similarly to the data relevance experiment in Appendix C.3, we compare each training result with a baseline. Here the baselines are the same as those reported in Section 5.1.2: for adversarial training we compare to the publicly available model of [56], and for stability training we use `Baseline_stab(50K)`.

We plot the results in Figure 14. As the figure shows, accuracy always grows with the data size, except for one errant data point at 40K unlabeled data with adversarial training, which performs worse than the baseline. While we haven't seen overfitting in this setting as we have when attempting

Figure 14: Performance improvement as a function of unlabeled data amount, relative to training without unlabeled data. When present, error bars indicate the range of variation (minimum to maximum) over 3 independent runs. **(a)** Difference in standard accuracy and in accuracy under the $\ell_\infty$ attack `PG_Ours`, for adversarially-trained models. **(b)** Difference in standard accuracy and in certified $\ell_2$ accuracy for smoothed stability-trained models, at different radii.

to reproduce [56], we suspect that the reason for the apparent drop in performance is that our training configuration was not well suited to so few unlabeled data. We also see in the plot that the higher the robustness, the larger the benefit from additional data.

The experiment shows that 100K unlabeled data come about halfway to achieving the gain of 500K unlabeled data. Moreover, for the most part the plots appear to be concave, suggesting that increase in data amount provides diminishing returns—even on a logarithmic scale. Extrapolating this trend to higher amounts of data would suggest we are likely to require very large amount of data to see another 7% improvement. However, the negative value at 40K unlabeled data hints at the danger of trying to extrapolate this figure—since we haven't carefully tuned the training at each data amount (including the one at 500K), we cannot describe any trend with confidence. At most, we can say that under our computation budget, model architecture and training configuration, it seems likely the benefit of unlabeled data plateaus at around 500K examples. It also seems likely that as computation capabilities increase and robust training improves, the point of diminishing returns will move towards higher data amounts.

### C.5 Effect of label amount

To complement the unlabeled data amount experiment of Appendix C.4, we study the effect of the *labeled* data amount. Here too, we only consider smaller amounts of labeled data than in our main experiment, since no additional CIFAR-10 labels are readily available. The main effect of removing some of the labels is that the accuracy of pseudo-labels decreases. However, since the main motivation behind robust self-training is that the pseudo-labels only need to be "good enough," we expect fewer labels to still allow significant gains from unlabeled data.

To test this prediction, we repeat the experiments of Section 5.1.2 with the following modification. For each desired label amount $n_0 \in \{2,4,8,16,32\}$K, we pick the first $n_0$ images from a random permutation of CIFAR-10 to be our labeled data, and consider an unlabeled dataset of $50\text{K} - n_0 + 500K$ images comprised of the remaining CIFAR-10 images and our 500K unlabeled images. We train a classification model on only the labeled subset, using the same configuration as in the "pseudo-label generation" paragraph of Appendix B.1, and apply that model on the unlabeled data to form pseudo-labels. We then repeat the final step of robust self-training using these pseudo-labels, with parameters as in Appendix B.1. Adding the remainder of CIFAR-10 without labels to our unlabeled data keeps the total dataset fixed. This allows us to isolate the effect of the quality of the pseudo-labels and hence the label amount on robust self-training.

Figure 15 summarizes the results of this experiment, which are consistent with our expectation that robust self-training remains effective even with a much smaller number of labels. In particular, for both adversarial robust self-training (Figure 15a) and stability-based certified robust self-training (Figure 15b), 8K labels combined with the unlabeled data allow us to obtain comparable robust accuracy to that of the state-of-the-art fully supervised method. Moreover, for adversarial-training we obtain robust accuracy only 2% lower than the supervised state-of-the-art with as few as 2K labels.

Figure 15: Performance for varying labeled data amount, compared to the fully supervised setting. **(a)** Standard accuracy and accuracy under the $\ell_\infty$ attack PG$_{0\text{urs}}$, for adversarially-trained models, as a function of label amount. The dotted line indicates the accuracy of the pseudo-label generation model, which we train with the labeled data only. **(b)** Certified robust accuracy as a function of $\ell_2$ radius for different label amounts, and our fully supervised baseline; RST$_{\text{stab}}(a+b)$ denotes the stability trained model with $a$ labeled data and $b$ unlabeled data, which consists of the $50\text{K}-a$ de-labeled CIFAR-10 images and our 500K unlabeled dataset.

In this small labeled data regime, we also see that the standard accuracy of the resulting robust model is slightly higher than the pseudo-label generator accuracy (the dotted black line in Figure 15a).

Two remarks are in order. First, in the low-label regime we can likely attain significantly better results by improving the accuracy of the pseudo-labels using standard semisupervised learning. The results in Figure 15 therefore constitute only a crude lower bound on the benefit of unlabeled data when fewer labels are available. Second, we note that the creation of our 500K unlabeled dataset involved a "data selection" classifier trained on all of the CIFAR-10 labels, and we did not account for that in the experiment above. Nevertheless, as the data selection model essentially simulates a situation where unlabeled data is generally relevant, we believe that our experiment faithfully represents the effect of label amount (mediated through pseudo-label quality) on robust self-training. Further, our experiments on the effect of the relevance of unlabeled data (described in Appendix C.3) suggest that using a slightly worse data selection model by training only on a subset of CIFAR-10 labels should not change results much.

### C.6 Standard self-training

We also test whether our unlabeled data can improve standard accuracy. We perform standard self-training to obtain the model SST(50K+500K); all training parameters are identical to RST$_{\text{stab}}$(50K+500K), except we do not add noise. SST(50K+500K) attains test accuracy 96.4%, a 0.4% improvement over standard supervised learning with identical training parameters on 50K labeled dataset. This difference is above the training variability of around 0.1% [54, 10], but approaches like aggressive data augmentation [10] provide much larger gains for the same model (to 97.4%).

### C.7 Performance against different PG attack radii

In Figure 16, we evaluate robustness of our state-of-the-art RST$_{\text{adv}}$(50K+500K) on a range of different values of $\epsilon$ and compare to TRADES [56], where we fine-tune attacks for each value of $\epsilon$ separately for each model (see Appendix B.3). We see a consistent gain in robustness across the different values of $\epsilon$.

## D SVHN experiment details

Here we give a detailed description on our SVHN experiments reported in Section 5.2.

### D.1 Experimental setup

Most of our experimental setup for SVHN is the same as in CIFAR-10, except for the following differences.

Figure 16: Comparing $\ell_\infty$ accuracy to tuned PG for various $\epsilon$. $\text{RST}_{\text{adv}}(50\text{K}+500\text{K})$ has higher accuracy than a state-of-the-art supervised model, across the range of $\epsilon$.

- **Architecture.** Throughout the SVHN experiments we use a slightly smaller Wide ResNet 16-8 model.

- **Robust self-training.** Since here the unlabeled data is 100% relevant, we set $w=1$ by sampling every batch from a single pool containing the labeled and unlabeled data. We still use $\beta = 6$ throughout; we performed brief tuning over $\beta$ to make sure that 6 is a reasonable value for SVHN.

- **Adversarial training.** We perform adversarial training with the same parameters as before and as in [56], with one exception: we set the perturbation magnitude $\epsilon = 4/255$ (we use step size 0.007 and 10 steps as in CIFAR-10). As we discuss below, adversarial training with the current configuration and $\epsilon = 8/255$ produced weak and inconsistent results.

- **Stability training.** No change.

- **Input normalization.** No change.

- **Data augmentation.** Following [54] we do not perform any data augmentation.

- **Optimizer configuration.** We use the same parameters as before, except here we use batch size 128.

- **Number of gradient steps.** We run 98K gradient steps in every SVHN experiment. For stability training we attempted to double the number of gradient steps and did not observe improved results.

### D.2 Pseudo-label generation and standard self-training.

To generate pseudo-labels we perform standard training of an Wide ResNet 16-8 model as described above on the core SVHN training set only. This model attains 96.6% test accuracy, and we use it to generate all the pseudo-labels in our SVHN experiments.

For comparison we repeat this procedure on the entire SVHN data (with all the labels). The resulting model has 98.2% test accuracy. Finally, we apply standard self-training using the configuration described above, i.e. we replace the SVHN extra labels with the pseudo-labels—this corrupts 1.6% of the extra labels (the extra data is easier to classify than the test set). Self-training produces a model with 97.1% accuracy, similar to the 0.4% improvement we observed on CIFAR-10, and 1.1% short of using true labels.

### D.3 Evaluation and attack details

We perform randomizes smoothing certification exactly as in the CIFAR-10 experiments in Section 5.1.2. For evaluating heuristic defenses, we fine-tune the PG attack to maximally break $\text{RST}_{\text{adv}}(73\text{K}+531\text{K})$ to obtain $\text{PG}_{\text{Ours}}$ with the following parameters: step-size $\eta=0.005$, number of steps $\tau=100$ and number of restarts $\rho=10$. We evaluate models at $\epsilon=4/255$, which is the same as the value we used during training.

Compared to CIFAR-10, we find that we require larger number of steps for SVHN attacks. Interestingly, for $\epsilon=8/255$ (which is what we evaluate our CIFAR-10 models on), we find that even after 1000 steps, we see a steady decrease in accuracy (when evaluated over a small random subset of the test set). However, for a smaller value of $\varepsilon=4/255$ (which is what we finally report on), we see that the accuracies seem to saturate after 100 steps of the attack.

| | $\ell_2$ radius: | 0 | 0.22 | 0.435 | 0.555 |
| | Enclosed $\ell_\infty$ radius: | 0 | 1/255 | 2/255 | 0.01 |
| Model | | | | | |
|---|---|---|---|---|---|
| `Baseline`$_\text{stab}$(604K) | | 93.6 | 84.9 | 70.0 | 59.8 |
| `RST`$_\text{stab}$(73K+531K) | | 93.2 | 84.5 | 69.7 | 59.5 |
| `Baseline`$_\text{stab}$(73K) | | 90.1 | 80.2 | 65.0 | 55.0 |

Table 6: SVHN certified test accuracy (%) for different $\ell_2$ perturbations radii, and the $\ell_\infty$ certified robustness they imply.

### D.4  Comparison with results in the literature

In the context of adversarial robustness, SVHN was not studied extensively in the literature, and in most cases there are no clear benchmarks to compare to. Unlike CIFAR-10, there is no agreed-upon benchmark perturbation radius for heuristic $\ell_\infty$ defenses. Moreover, we are not aware of a heuristic SVHN defense that withstood significant scrutiny. A previous heuristic defense [20] against attacks with $\epsilon = 12/255$ was subsequently broken [14]. In [41], the authors study SVHN attacks with $\epsilon = 4/255$ but do not tabulate their results. Visual inspection of their figures indicates that we get better robust accuracies than [41] by over 7%, likely due to using a higher capacity model, better training objectives and better hyperparameters.

Two recent works constitute the state-of-the-art for certified robustness in SVHN. Cohen et al. [9] study randomized smoothing certification of $\ell_2$ robustness and report some results for SVHN, but do not tabulate them and did not release a model. Their figure shows a sharp cutoff at radius 0.3, suggesting a different input normalization than the one we used. In view of our comparison in Appendix B.5, it seems likely that our model attains higher certified accuracy. Gowal et al. [17] propose interval bound propagation for certified $\ell_\infty$ robustness. They report a model with 85.2% standard accuracy, and 62.4% certified robust accuracy against $\ell_\infty$ attacks with $\epsilon = 0.01$.

In Table 6 we list selected point from Figure 3, showing certified accuracy as a function of $\ell_2$ perturbation radius. For each $\ell_2$ radius we also list the radius of the largest $\ell_\infty$ ball contained within in it, allowing comparison between our results and [17]. At the $\ell_2$ radius that contains an $\ell_\infty$ ball of radius 0.01 we certify accuracy of 59.8%, less than 3% below than the result of Gowal et al. [17]. This number is likely easy to improve by tuning $\sigma$ and $\beta$ used in stability training, situating it as a viable alternative to interval bound propagation in SVHN as well as CIFAR-10.

## E  Comparison to Uesato et al. [48]

Independently from our work, Uesato et al. [48] also study semisupervised adversarial learning theoretically in the Gaussian model of [41] and empirically via experiments on CIFAR-10, Tiny Images, and SVHN. Overall, Uesato et al. [48] reach conclusions similar to ours. Here, we summarize the main differences between our works.

We can understand the algorithms that [48] propose as instances of Meta-Algorithm 1 with different choices of $L_\text{robust}$. In particular, their most successful algorithm (UAT++) corresponds to

$$L_\text{robust}^\text{UAT++}(\theta,x,y) = \max_{x' \in \mathcal{B}_\epsilon^p(x)} L_\text{standard}(\theta,x',y) + \lambda L_\text{reg}(\theta,x),$$

where $L_\text{reg}(\theta,x) = \max_{x' \in \mathcal{B}_\epsilon^p(x)} D_\text{KL}(p_\theta(\cdot \,|\, x) \,\|\, p_\theta(\cdot \,|\, x'))$ as in (6). In contrast, we do not maximize over $L_\text{standard}$, i.e. we use

$$L_\text{robust}(\theta,x,y) = L_\text{standard}(\theta,x,y) + \lambda L_\text{reg}(\theta,x).$$

Both this work and [48] perform adversarial training on CIFAR-10 with additional unlabeled data from Tiny Images. Both works consider the benchmark of $\ell_\infty$ perturbations with radius $\epsilon = 8/255$ and report results against a range of attacks, which in both papers includes `PG`$_\text{TRADES}$ [56]. For this attack, our best-performing models have robust accuracies within 1% of each other (their WRN-106-8 is 1.1% higher than our WRN-28-10, and their WRN-34-8 is 1% lower), and we obtain about 3% higher standard accuracy.

Beyond the algorithmic and model size difference, Uesato et al. [48] use different training hyperparameters, most notably larger batch sizes. Additionally, to source unlabeled data from 80 Million Tiny images they use a combination of keyword filtering and predictions from a standard CIFAR-10 model. In contrast, we do not use keywords at all and train a classifier to distinguish CIFAR-10 from Tiny Images. Both works remove the CIFAR-10 dataset prior to selecting images from TI; we also remove an $\ell_2$ ball around the CIFAR-10 test set. Due to these multiple differences and similar final accuracies, we cannot determine which robust loss provides better performance.

Uesato et al. [48] perform a number of experiments that complement ours. First, they show strong improvements in the low labeled data regimes by removing most labels from CIFAR-10 and SVHN (similar findings appear also in [55, 32]). Second, they demonstrate that their method is tolerant to inaccurate pseudo-labels via a controlled study. Finally, they propose a new "MultiTargeted" attack that reduces the reported accuracies of the state-of-the-art robust models by 3-8%. Contributions unique to our work include showing that unlabeled data improves certified robustness via randomized smoothing and studying the effect of irrelevant data theoretically and experimentally.

## Footnotes

[2] Zhang et al. [56] write the regularization term $D_{\text{KL}}(p_\theta(\cdot \mid x') \| p_\theta(\cdot \mid x))$, i.e. with $p_\theta(\cdot \mid x')$ rather than $p_\theta(\cdot \mid x)$ taking role of the label, but their open source implementation follows (6).

[3]We exclude any image close to the CIFAR-10 test set; see Appendix B.6 for detail.

[4]`https://github.com/lyakaap/VAT-pytorch`

[5] https://github.com/uoguelph-mlrg/Cutout

[6] https://github.com/DeepVoltaire/AutoAugment