[Reviews · NeurIPS 2019]

Reviewer 1



Originality - it's the first time that semi-supervised techniques are shown to effective in the robust learning regime, both theoretically and empirically. Significance - robust models require more samples in order to generalize. Showing that unlabeled data alleviate this problem is crucial because it is much easier (and cheaper) to collect. Quality and Clarity - the paper is well written. The claims and methods are clearly explained.

Reviewer 2



Originality: Previous theoretical work on the subject is to quantify the amount of additional labeled data required to attain non trivial robust error whereas Theorem 2 quantifies the additional unlabeled data required. The contribution due to the meta algorithm is minor since classification with L^stab and L^adv has been studied before. Quality: The theoretical results are sound and the claims are well supported from the experiments Clarity: The paper is well written for the most part. The term certified l2/accuracy is not defined. Line 94 says difficult to learn classifier. For the gaussian model, the classifier must be easy to learn. Isn’t that so? Line 126-128. It is difficult to follow the logic in line “As n grows … goes to 0”. It seems that it is based some unexplained geometry. Same goes for the line with “parameter scaling”. Significance: The theoretical results are for a simple gaussian model, instead of a more realistic one. The results on real datasets might be quite different. Using more datasets for experiments might be more convincing. Furthermore the ratio of positives to negatives 1, which is again a special case. What happens when there is class imbalance?

Reviewer 3



This paper theoretically and empirically shows that guarantee of non-trivial adversarial robustness only requires more unlabeled data. Strengths: 1. The paper theoretically proves that under the Gaussian model, more unlabeled data is enough to certify small robust accuracy (1e-3 in the paper) by their robust self-training algorithm. 2. The paper also empirically shows on cifar10, robust self-training algorithm with unlabeled data can outperform state-of-art models and standard self-training. 3. The paper empirically illustrates on SVHN that robust self-training with unlabeled data almost achieves the same robust accuracy as the robust training with labeled data. 4. The paper is clearly written. It is a pleasure to read it. Weakness: 1. The main concern is that the connection between the theory and the experiment is loose. The theory has very strong assumptions on the true model (Gaussian model). This is totally different from the real world dataset model like cifar10 and SVHN. The authors never addresses the connection anywhere in the paper. Theoretical guarantee for the real world data still remains an open question. 2. The comparison seems to be unfair with the state of art models because robust-self training has extra unlabeled data information. Some empirical analysis of state-of-art model utilizing unlabeled data can be interesting. Minors: 1. Why the state of art model for l_inf attack is different for epsilon = 2/255 and 8/255? Does that mean state-of-art model can only guarantee one specific epsilon? 2. RST standard accuracy 80.7 (Figure 1 b) when epsilon = 2/255 is much lower than standard accuracy 89.7 when epsilon (Table 1). Why is that? Training with small epsilon intuitively should give higher standard accuracy. -------------------------------------------------- Updates after rebuttal -------------------------------------------------- Thanks the authors for providing a super clear rebuttal. My questions are addressed.

[Author Response · NeurIPS 2019]

We thank the reviewers for the kind and helpful reviews. Below we address each review in turn.

**Reviewer 1** We thank the reviewer for recognizing the novelty and significance of our
results, as well as for the additional important references about robust generalization—
we will include them in the revised paper. We also thank the reviewer for the excellent
suggestion to study how many *labels* are actually necessary for achieving state-of-the-art
robustness. Following this suggestion, we perform robust self-training with random subsets
of CIFAR-10 as the labeled data and the remainder of CIFAR-10 and our mined 500K
images from Tiny Images as unlabeled data. The figure to the right shows the result (for
adversarial training and testing with $\mathrm{PG}_{\mathtt{Ours}}$ with $\epsilon = 8/255$): with as few as 4K labels, we
are able to match the fully-supervised state-of-the-art! The revised paper will include a
detailed account of this experiment.

**Reviewer 2** We thank the reviewer for the valuable feedback, which will improve the readability of our paper. Below,
we address each point in the review; we will also carefully revise our paper to clarify each of these points.

*"The term $\ell_2$ certified accuracy is not defined."* By certified $\ell_2$ accuracy $\xi$, we mean a proof that $\mathrm{err}_{\mathrm{robust}}^{2,\epsilon}$ defined in Eq. (2)
is at most $1 - \xi$ on the test set. Specifically, we use the randomized smoothing proof by Cohen et al. (2019).

*"Line 94 says difficult to learn classifier. For the Gaussian model, the classifier must be easy to learn. Isn't that so?"*
Here by "difficult to learn" we meant "requires many samples to learn."

*"Line 126-128. It is difficult to follow the logic..."* Here is a more detailed explanation: we have $\hat{\theta}_{\mathrm{final}} = (\frac{1}{\tilde{n}} \sum_{i=1}^{\tilde{n}} \tilde{y}_i y_i)\mu +$
$\frac{1}{\tilde{n}} \sum_{i=1}^{\tilde{n}} \tilde{y}_i \varepsilon_i$ where $\varepsilon_i \sim \mathcal{N}(0, \sigma^2 I)$ is the noise in example $i$. In the proof we show that with high probability
$\frac{1}{\tilde{n}} \sum_{i=1}^{\tilde{n}} \tilde{y}_i y_i \geq \frac{1}{6}$ while the variance of $\frac{1}{\tilde{n}} \sum_{i=1}^{\tilde{n}} \tilde{y}_i \varepsilon_i$ goes to zero as $\tilde{n}$ grows, and therefore the angle between $\hat{\theta}_{\mathrm{final}}$ and
$\mu$ goes to zero. By Eq. (11) in Appendix A.1 and Eq. (3) this implies that the robust error becomes small.

*"Using more datasets for experiments might be more convincing."* We agree and will gladly experiment on more datasets.
Unfortunately, there are not many established benchmarks for adversarial robustness (we do not have the computational
resources for adversarial training on ImageNet). We would appreciate suggestions for additional datasets to consider.

*"What happens when there is class imbalance?"* The theoretical results in this work easily extend to the case where there
is class imbalance. The upper bounds in Proposition 1 and Theorem 2 hold regardless of the label distribution. The
lower bound in Theorem 1 changes from $\frac{1}{2}(1 - d^{-1})$ to $p(1 - d^{-1})$ where $p$ is the proportion of the smaller class; the
only change to the proof in [34] is a modification of the lower bound on $\Psi$ in page 29. We thank the reviewer for raising
this interesting question—we will mention it in the revised paper.

**Reviewer 3** Thanks for the helpful comments (addressed below) and for finding our paper a pleasure to read.

*"The main concern is that the connection between the theory and the experiment is loose."* We believe that there is a
substantial connection between theory and experiment in our paper because both parts follow the same algorithmic
approach. As we mention in lines 288–289, this is not a coincidence: our theoretical results motivated our experimental
investigation. In particular, the observation that self-training is very effective in utilizing unlabeled data in the Gaussian
model led us to empirically test it in more realistic settings. We agree that showing a sample complexity separation in a
more realistic model is a challenging open problem. We thank the reviewer for pointing out that this connection wasn't
clear enough in our paper; we will state it clearly in the introduction to the revised paper.

*"The comparison seems to be unfair with the state of art models because robust self-training has extra unlabeled data
information."* Our main contribution is to show that unlabeled data improves adversarial robustness, and therefore our
primary experiments focus on evaluating this improvement using state-of-the-art models. However, in Table 1 we do
compare against [15] that uses additional labeled data from ImageNet. Since we are the first to propose semisupervised
learning for adversarial robustness, there were no previous methods for using unlabeled data that we could directly
compare to. Nevertheless, in Appendix C.1 (described in lines 229–234), we compare our proposed method (RST) to a
state-of-the-art method for standard semisupervised learning (VAT), which we adapt to the robustness setting. In this
comparison, both methods use the same unlabeled data, and we find that RST offers significantly stronger performance.

*"Minors."* The reviewer raises questions about the relation between results in Table 1 and Figure 1b, both of which report
robustness to $\ell_\infty$ attacks but differ in a crucial aspect: Table 1 shows results for *heuristic defenses* tested against strong
gradient-based attacks (with $\epsilon = 8/255$), while Figure 1b compares methods with *certified (proven) robustness* against
*all attacks* with (with $\epsilon = 2/255$). State-of-the-art heuristic defenses utilize different algorithms than state-of-the-art
certified defenses and therefore we cite different works in each case. Moreover, since proving robustness to all attacks is
a harder problem than defending against particular attacks, the former comes at a cost to standard accuracy, explaining
the difference the reviewer points out.

[Meta-Review · NeurIPS 2019]

This paper presents a theoretical analysis on using unlabeled data (under a self-training scheme of a Gaussian model) to improve the robustness against adversarial noise, followed by a semi-supervised learning method to learn deep networks. The empirical results are state-of-the-art. However, this paper heavily overlaps with another paper "Are Labels Required for Improving Adversarial Robustness?". As a condition to accepting and including the paper in the proceedings, put the following disclaimer in the footnote on the first page: "The authors declare that the present paper is independent of "Are Labels Required for Improving Adversarial Robustness?"."